# ¿Dónde Vive la Ciencia en su Comunidad?: How a Community Is Using Photovoice to Reclaim Local Green Spaces

Espacio: Familias y Comunidad [1,2,†]

1    Straus Center for Young Children & Families, Bank Street College of Education, 610 W 112th Street, New York, NY 10025, USA; cmedellin@bankstreet.edu or dballesteros@nysci.org
2    New York Hall of Science, 47-01 111th Street, Corona, NY 11368, USA
†    A full author list is available in Appendix A.

**Abstract:** The *¿Dónde Vive la Ciencia en su Comunidad*? (where does science live in your community?) photovoice project is a community-based participatory research project that investigates the presence and influence of science within local environments. In collaboration with researchers, science, technology, engineering, mathematics (STEM) educators, and community members from the Latine community in Corona, Queens, the project investigated where science is found in our communities. Community researchers used photography to document their surroundings and identified key themes related to the role of science through technology, community health, safety, and wellness. The photovoice method elevated social justice issues through critical dialog, creating opportunities for change through collective action. Among the critical issues discussed were urban planning, specifically the impacts of gentrification on the local community and the possibilities that greening offered as a site of agency, multigenerational learning, and resistance through ways of knowing. Community researchers examined the dual nature of STEM as both a tool of control and a means for justice, interrogating whose voices and experiences are prioritized in decision-making processes. Establishing shared green spaces emerged as an act of epistemic disobedience and resistance for sustaining community health and cultural identity. The project highlights how collaborative, community-led initiatives promote the reclamation of political power through collective action and disrupt colonizing forces, offering actionable recommendations for policy, research, and practice to guide justice-oriented change.

**Keywords:** photovoice; community-based participatory research (CBPR); participatory action research (PAR); green spaces; community-driven urban planning; epistemic disobedience; buen vivir (good living)

## 1. Introduction

### 1.1. An Invitation to Our Readers

We begin this article with an invitation to you, dear reader, to witness our collective process of epistemic disobedience through community-engaged research. Here, we refer to epistemic disobedience as a deliberate shift away from institutionalized, Western ways of knowing. Our collective research group, *Espacio: Familias y Comunidad* (Space: Families and Community), is rooted in interconnectedness. We come together as immigrants, children of (im)migrants, first-generation scholars, academically (and colonially)-trained researchers, community researchers, community leaders, advocates, mothers, daughters, sons, and friends. Our identities are genderfluid, multiracial, and multigenerational, spanning Latina,

Chicana, Colombian, Ashkenazi Jewish, White, Ecuadorian, Chinese, Cuban, Hawai'ian, and Mexican heritages. We are students, educators, social workers, public health advocates, museum professionals, and volunteers who share a deep commitment to the community.

Embracing Jiménez Estrada's Tree of Life as research methodology (2005), we recognize that we each walk our own paths, with our own ways of knowing and being. From our unique vantage points, we have cultivated a shared vision of science's role in community wellbeing and have turned our attention to the green spaces that nurture us. Like the tree of life—the ceiba—that serves as a sacred symbol in Maya cosmology, we honor the balance between our individual histories and our shared purpose and understand that our actions today are informed by our individual and collective histories and echo into the future.

We recognize our shifting roles as insiders and outsiders—an important reminder to regularly interrogate the role that power plays in our work. Our project, ¿Dónde vive la ciencia en su comunidad?, reflects the diversity of our group, and is rooted in everyday life. Science, like the tree of life, thrives in the relationships between people, the environments we inhabit, and the stories we tell. Together, we nurture this inquiry with respect, reciprocity, and a shared responsibility for the future, understanding that the seeds we plant today will serve to strengthen the roots of our communities. We take this time to share a bit about who we are in order to invite[1] you to join us in a space at the borders, *nepantla*, where we embrace diversity in ways of knowing and being and lean into epistemic friction (Medina 2013) or *choques* (crashes) (Torre and Ayala 2009) as we explore the complexities of community-based research.

*1.2. Where Science Lives in Our Community*

Through this project, we recognize the power in local knowledge, with a focus on lived experiences, culture, and spirituality, especially from communities that have historically been erased or marginalized through knowledge production (Jolivétte 2015). Mignolo (2011) describes epistemic disobedience as an active "de-linking" from, and complicating of, Western ways of knowing. Epistemic disobedience leads us to decolonial options that challenge colonial logics of domination and power (Quijano 2000) and affirm onto-epistemologies outside European genealogies: "Epistemic disobedience takes us to a different place, to a different 'beginning' (not in Greece, but in the responses to the 'conquest and colonization' of America and the massive trade of enslaved Africans), to spatial sites of struggles and building rather than to a new temporality within the same space. . ." (p. 45). De-linking from colonial logics aligns with the work of womanist scholars like AnaLouise Keating (2013) and Jillian Ford (2023) who advocate for relationality as a strategy of critical consciousness, with an emphasis on the role of authenticity (and at times, vulnerability). It is through the practice of invitation, and the act of "listening with raw openness" (Keating 2013, p. 197) that we open ourselves to the possibilities that emerge from radical connection.

This collaborative community-based participatory research (CBPR) project brought together members of the Latine[2] community of Corona-Flushing, Queens, and members of two educational institutions, Bank Street College of Education (BSC) and the New York Hall of Science (NYSCI). The partnership explored where and how the local community perceived and experienced science in their everyday lives through participatory methods such as picture taking and storytelling. BSC is a graduate school in New York City with a rich history in preparing educators to challenge critical thought through progressive teacher education and research. NYSCI is a science museum nestled in the heart of a vibrant neighborhood with rich histories of immigration in the borough, Queens, New York. The two educational institutions shared a common value—placing community at the core of knowledge creation processes and a shared interest in conducting participatory forms of research that center the voices of community members, in order to create opportunities

for more equitable and community-driven forms of science, technology, engineering, and mathematics (STEM) learning.

Building on this mutual interest, the project was conceptualized to investigate science learning through inquiry, asking the question, "where does science live in our daily lives and local community?" The research question was intentionally broad and inclusive—an invitation to community members to share their own experiences with science across a broad spectrum. Staff from both institutions shared information about the goals of the project with community members who visited NYSCI and encouraged them to invite other friends and family to participate as well. Through this process, we established our collaborative group, Espacio: Familias y Comunidad (Espacio), which included staff from NYSCI and BSC, and 15 residents of Corona, Elmhurst, and Flushing. Together, the Espacio project team focused on the implications of science on community health, safety, and wellbeing. By involving community members as researchers, this project challenged traditional research paradigms that often overlook the voices and experiences of those affected by urban planning and policy decisions.

The intersection of science and community wellbeing has increasingly become a focal point of research, especially in the context of environmental justice. Urban settings are of particular interest, where rapid changes to the natural and/or built environment can both empower and marginalize local populations. The current project investigated this dual nature of science in urban environments. On the one hand, advancements in science can contribute to improved living conditions and greater access to resources. At the same time, science can also exacerbate existing inequalities, particularly when used in ways that prioritize power preservation and the interests of wealthier populations over those of local communities. This tension is evident in the phenomena of gentrification, specifically green gentrification, where seemingly eco-conscious policies and technologies often result in the displacement of vulnerabilized[3] populations rather than benefiting them (Jelks et al. 2021).

Within Black, Indigenous, people of color (BIPOC), and immigrant communities, these changes in urban environments often create or amplify socioeconomic disparities and historical patterns of exclusion from decision-making processes (Schell et al. 2020). Understanding how local communities perceive and engage with science in their daily lives is crucial for developing inclusive and culturally responsive policies and programs across various socioeconomic and political levels (interpersonally, institutionally, and at the local and state levels governments). Providing multi-level opportunities for engagement is necessary for developing asset-rich narratives of local communities so that decision makers and policies can be designed based on community strengths and assets rather than risks.

In this research project, we aimed to explore these issues in partnership with members of a Latine community to avoid replicating exclusionary practices and inequitable structures within the research process. By centering the voices, experiences, and knowledge of local community members, we aimed to contribute to a more nuanced understanding of how science shows up and is experienced by Latine individuals and families in urban settings. Through the process of co-learning, this research offers recommendations for researchers, museum educators, community leaders, and policymakers—to not only understand how Latine communities experience science but to also develop actionable recommendations to address (and in some cases, dismantle) inequitable uses of science and envision more equitable possibilities.

Members from both Institutions acknowledged their positionality and power, and recognized that as academics, they may inadvertently participate in "academic capitalism" (Parra 2022), which prioritizes epistemic overproduction, extraction, exploitation, and violence (Pérez 2022). Author, educator, and activist bell hooks (1994) observed how the academy has historically appropriated the work of women of color in order to reproduce

intellectual hierarchies in which certain (read: Eurocentric, colonial, highly abstract) epistemes are deemed superior to others (e.g., Indigenous worldviews). The theories that are privileged in the academy, and particularly in the sciences, are often inaccessible to the public, further exacerbating class divides. In an effort to resist these cycles of epistemic injustice and devaluation, the staff at BSC and NYSCI surrendered the "official" title of researcher and centered community members as the researchers (Jolivétte 2015). According to Jolivétte (2015) and the research justice framework, when we approach research by prioritizing our relationships to one another as sacred, we begin to shift power dynamics:

> When we redefine methodologies within the context of the sacred, we shift the fundamental relationship of the research process from one based upon unequal power relationships to one based upon mutual respect and reverence for all those impacted by the focus of our studies, documentation, and efforts to reform policy. (p. 7).

The staff at BSC and NYSCI lovingly identify as *compas*[4] in this work: colleagues, co-conspirators, allies, and advocates who aim to redefine and resist traditional hierarchical structures within research and the academy. The compas served as companions and witnesses in their community's research journey.[5]

Photovoice, a participatory action research (PAR) method, offers a unique approach to engaging local community members in the knowledge production process to effect social change through photography and storytelling. In photovoice studies, community members are supported to take on the role of researchers through in-depth training as they engage in data collection, documentation, and interpretation. This method not only captures local knowledge but also serves as an entry point for collective critical analysis and dialog. Furthermore, the photovoice method creates a shared language through photographic displays in order to catalyze advocacy and collective action, allowing researchers to make connections between their personal experiences and broader social change. Previous studies have demonstrated the effectiveness of photovoice in addressing public health issues and promoting community resilience, particularly among historically vulnerabilized groups (Hergenrather et al. 2009; Seitz and Orsini 2022; Strack et al. 2022; Suprapto et al. 2020). However, its application in exploring the role of science in urban Latine communities remains underexplored. Latine communities in New York City are positioned in ways that complicate discourse around the distinctions between Western and Indigenous onto-epistemologies, shaped by their immigration histories, their non-native status on the lands they inhabit, and their continued subjection to inequitable STEM systems rooted in colonial and modernist logics. The current project highlights the ways that the photovoice process created space for community members to resist urban planning decisions and reinterpret the role of science in their lives. A central hypothesis of this work is that community-driven approaches to urban planning and environmental stewardship can serve as powerful decolonial strategies that promote community health and wellbeing, cultural identity, and social justice.

### 1.3. Theoretical Frameworks

The group's work together was guided by theoretical approaches that focus on redistributing power and working towards social justice. Freire's critical consciousness (*conscientização*) provided a lens for exploring how community researchers documented and understood our community's assets and limitations. Critical consciousness involves becoming aware of social, political, and economic contradictions and taking action against oppressive elements of reality (Freire [1970] 2000). For this particular project, this critical consciousness raising was a collective experience.

Feminist and womanist approaches were also central to our work. The project embodied popular feminist education, which Doerge (1992) describes as "transforming the world where women stand" (p. 5). In this view, political change and social transformation should be guided by the experiences of the *cotidiano* [the everyday], especially the traditions and ways of knowing that may be excluded from dominant paradigms. A majority of the community researchers in our group identify as women, and gender plays an important role in our collective ability to transform local environments. Environmental justice research highlights that women are vulnerable to natural disasters, food insecurity, and violence (Correa 2022). Nevertheless, gender inequality leads to women being exploited and undervalued in environmental decision making (Bell 2016). The effects of gender inequality on women are also influenced by race, sexual orientation, legal status, and socioeconomic class, highlighting the need for an intersectional approach to environmental justice. For these reasons, feminist theory and a womanist orientation were necessary for understanding the impacts of science on our local community. Through feminist popular education, and with a focus on the cotidiano, the project created a space where women played an active role as knowledge producers, allowing new paradigms to be explored and developed. As one Espacio member stated:

> *Me emociono porque como muchas madres ... [ustedes] ya saben más o menos cual es la labor de una mujer. Es mucho trabajo, ¿verdad? Y pues agradezco que tuve la oportunidad de aprender. . .de poder hablar como dicen. . .aprendemos nosotros junto a los que han estudiado más que uno [I get emotional because many mothers. . .you already know more or less what is the labor of a woman. It's a lot of work, right? And well I am grateful that I had the opportunity to learn. . .to be able to talk like they say. We learn together with those who have studied more than us]* (from BSC Alumni Event).

Our methodological and analytical approaches are built on these theoretical foundations.

The photovoice process was developed by researchers Wang and Burris (1994; 1997) to engage rural Chinese women in reshaping their local policies by documenting positive and negative aspects of their daily lives and local contexts. The method is rooted in both critical consciousness and feminist theory and has been widely utilized in public and environmental health research to address issues concerning food accessibility, pollution, and urban planning (Brandt et al. 2017; Carpenter 2022; Gahan et al. 2022). These studies typically use photovoice to work towards social and environmental justice for disenfranchised communities by sharing research findings with other local community residents, researchers, and policymakers. Furthermore, photovoice has become more popular and accessible as a research method in recent decades, and several studies have utilized it with communities from diverse racial, ethnic, gender, and socioeconomic backgrounds (Carroll et al. 2018; Wendel et al. 2019).

*Pláticas*—a communal act of storytelling grounded in a *feminista* framework— created a context for collective sharing and authentic conversations (Fierros and Delgado Bernal 2016; Morales et al. 2023). Storytelling through pláticas, and the act of re-membering were central to disrupting traditional power structures and expanding how knowledge production took place within the Espacio group. Pláticas are based on conversations between Latine family members to share stories, lessons, and cultural knowledge with one another (Fierros and Delgado Bernal 2016). In our meetings, they allowed community researchers to share their personal stories with one another in an intimate setting while also cultivating a healing experience by allowing them to express their emotions when sharing their positive and negative experiences with science in their community (Fierros and Delgado Bernal 2016; Flores Carmona et al. 2018). These methods highlight the transformative potential of feminist popular education and feminista frameworks and also

make a case for the necessity of creating intentional *espacios* [spaces] where women can contribute to the narratives surrounding science and environmental justice.

Embracing a womanist perspective and the feminista approach of plática served for us as a practice in epistemic disobedience—a concept grounded in decolonial frameworks (Mignolo 2011). The coloniality of knowledge (Quijano 2000) is pervasive and influences the structures that directly influence our day to day lives. Mignolo called for an ideological "delinking" from colonial logics and describes epistemic disobedience as an act that challenges dominant onto-epistemologies and simultaneously validates and uplifts local ways of knowing and being. By centering plática—a culturally rooted form of collective narrative and reflection—our project sought to disrupt the Western epistemologies that govern scientific knowledge creation. As a discipline, science has historically excluded BIPOC and, specifically, women. Plática became an enactment of epistemic disobedience, where our understandings of science and our knowledge creation around it were returned to the community. It was through our collective epistemic disobedience that we, as a collective, disrupted our own assumptions about Latine women's connection with science, and through this knowledge-building process, we aimed to counteract the historic and systemic colonial structures that exclude Latine women from science education and research.

While some scholarship might frame our efforts as "decolonizing", we have intentionally chosen to avoid this terminology. Following Tuck and Yang's (2012) lead through their influential critique, we acknowledge that decolonization speaks to the specific struggles of Indigenous sovereignty, land rights, and self-determination. In our context, we use terms like decoloniality, epistemic disobedience, epistemic justice, and community-centered knowledge production to reflect the project's goals of redistributing epistemic power, affirming local knowledge, collective action, and reclaiming political power. We refrain from describing this project as "decolonizing" and instead focus on describing the processes that have helped us move towards epistemic justice. Specifically, we identify relationality, remembering, and the co-creation of knowledge through plática as inherently and epistemically disobedient.

In conceptualizing the role of NYSCI as a site for the ongoing discussions in this project, we were guided by Gloria Anzaldúa's interpretation of the concept of *nepantla* (2015) (from the Nahuatl word "in between"). Cultural sites like NYSCI can facilitate the experience of nepantla, a liminal space where contradictions can coexist and where engaging *nos/otras* (us/them; insider/outsider) means new understandings and transformation become possible (Torre and Ayala 2009). In this way, Anzaldúa's (2015) notion of the *nepantlera* (or the one who navigates between worlds) offers a powerful metaphor for the roles that our collective group, Espacio, took on for this project. Through our work, we bridge multiple sites, positionalities, and experiences within local communities in order to envision opportunities for collective action and change. This perspective aligns with the Indigenous worldview of *buen vivir* (good living), which emphasizes the importance of harmony with nature, community wellbeing, and the collective over the individual (Acosta and Martínez Abarca 2001). An orientation towards buen vivir challenges conventional approaches to urban development that often marginalize and even criminalize immigrant communities, proposing instead development that is inclusive, equitable, and community-driven. This participatory project embodied the principles of buen vivir by elevating the voices of community members in examining the impact of science on individual and collective experiences, and in identifying opportunities for taking collective action to cultivate community wellbeing.

Finally, the project redefined power by deepening participatory and reflexive practices. Reflexivity was crucial in this project, as it created pathways for the community members to exercise collective agency and tell their stories without fear or judgment. The compas

were also aware of the potential limitations of photovoice as a method, as the scientific rigor and unequal power dynamics between them and community members might limit the extent to which the latter is truly representing their own ideas and involved in shaping their own outcomes (Ng et al. 2023).

Like the stories and understandings that were quilted together through the community researchers' photographs, the weaving of these theoretical and methodological strands serves as an offering to educators, researchers, and policymakers for strategies around community engagement. The concept of buen vivir and how feminista framing applies to our focus on reshaping and reclaiming local green spaces for social/communal health and family wellbeing (also for educational opportunities).

## 2. Materials and Methods

### 2.1. Community Researchers/Compas and Study Design

As of 2022, the New York City borough of Queens is home to over 2.3 million New Yorkers with over 46 percent being born outside of the U.S. (Data USA n.d.). The borough's diverse population largely consists of Hispanic/Latine, Asian, and Black communities and is recognized for its rich culture, food, and linguistic diversity. Queens has recently become known for its resilience after becoming an epicenter of the COVID-19 pandemic that affected millions of community members, especially within majority immigrant neighborhoods such as Elmhurst and Corona (NYC Mayor's Office of Immigrant Affairs 2020; Kimiagar et al. 2019). Residents of these neighborhoods described how their vibrant community became a silent battleground for economic, social, and emotional survival due to unemployment, severe rent burden, social isolation, and restricted access to adequate health and social care services (Correal and Blue 2021). In May 2020, the Elmhurst and Corona Recovery Collaborative (ECRC)—a coalition of more than 20 local community organizations—responded to local residents' needs by providing food, personal protective equipment (PPE), economic assistance, and health and social services. As a member of the ECRC, NYSCI responded to the educational and social needs of families in Elmhurst and Corona through regular community engagement events and by establishing the Family Advisory Committee (FAC), an annual cohort of 10 community members who meet with the museum staff and share their ideas and feedback on existing and future family engagement programming, community celebrations, and museum exhibits.

#### 2.1.1. Recruitment

Community researchers were recruited using a combination of direct outreach and snowball sampling. Initial community researchers were identified through the Family Advisory Committee (FAC) at NYSCI. Recruitment efforts also included outreach during a Family Day event at NYSCI in June 2023, and leveraging strategies effective in the community, such as word-of-mouth and in-person communications. A total of 15 community researchers and their families were recruited from the Corona, Jackson Heights, and Elmhurst communities, many of whom had previous relationships with the museum and the project research team. In addition to their previous engagement with NYSCI through various programs, the recruited community researchers were active members of their community prior to their participation in this project, often participating in various community-based groups and events held by other community organizations, including initiatives around voting, parent–teacher associations, and parental literacy awareness. All 15 community researchers and their families completed the project's initial six sessions and exhibitions of the group's work. Although the number of community researchers was small, they were selected to represent a diversity of perspectives within the community. This allowed for in-depth exploration of the research question.

### 2.1.2. Ethical Considerations

Ethical considerations were integral to the project, particularly in terms of accessibility and language justice. Recognizing the linguistic needs of the Espacio community, meetings were conducted primarily in Spanish and were led by bilingual researchers. This approach aimed to prevent linguicism and ensure that all community researchers could fully participate in the project (Antena Aire 2020; Doan et al. 2024). Linguistic dominance or linguicism is the idea that certain languages hold more value and are more respected than others (Antena Aire 2020). Leading studies solely in English or providing ineffective interpretation services can uphold social disparities that limit community partners' voices and decision-making power (Doan et al. 2024). Language accessibility is pivotal when researchers work with linguistically diverse communities. Researchers have historically perpetuated linguicism by leading recruitment, co-design, and dissemination solely in English, leaving community members in the dark about the project behind the scenes. By using a language justice framework for this project, we recognize the importance of effectively communicating project results, updates, and next steps with community researchers. This approach also meant that although most compas were involved directly in the photovoice process, those who identified as Latine and spoke Spanish fluently were more likely to communicate directly with the community researchers. Qualitative researchers such as Dwyer and Buckle (2009) highlighted this occurrence as the insider and outsider dynamic, where insiders are considered to be better able to work with a certain group because they belong to the community involved, while outsiders are not. Rather than resolve this tension, we centered the ideas of the community researchers by designing meetings with their needs and ways of knowing at the center.

### 2.1.3. Photovoice Methodology and Session Structure

The collaborative project was facilitated in Spanish; however, in keeping with language justice principles, we ensured that the group's entire language repertoire was honored, moving between English and Spanish as needed (Doan et al. 2024). The project consisted of six sessions, two exhibitions, and five pláticas—*cafecitos* (coffee hours)—which served as a time to eat, gather, reflect, and learn with each other. The photovoice sessions were hosted monthly at NYSCI. Each of the sessions lasted three hours, with at least 60% of the community researchers attending in person at each session. Weekly planning meetings were held at the museum to discuss session agendas, objectives, and necessary materials. The sessions were guided by the Photovoice Path (Lorenz 2005), which facilitated the collection, analysis, and interpretation of data. The Photovoice Path was adapted for cultural relevance and to honor the iterative, non-linear nature of this project, and specifically pláticas were integrated into the session structures (see Figure 1).

Initial sessions were dedicated to reviewing the photovoice process, including establishing community norms for discussion sessions and an introduction to ethical research practices (i.e., the role of informed consent when photographing subjects). Community researchers learned about the photovoice process by reviewing sample photos captured by the compas prior to the initial kick-off meeting. These photos are examples of where science can appear in everyday life within the original photographer's community such as a person recycling plastic bottles, solar panels on streets, and a close-up of a tree. Community researchers reviewed these photos and wrote their own descriptions of what they observed. They then shared what they found interesting about their set of photos in their small groups. Community researchers compared their interpretations of the images to the captions written by the original photographers to learn about the rich stories behind them. This exercise allowed the whole group to understand how photos can harbor broader messages when

context and narrative are added. Some of these messages can reflect satisfaction with community offerings while others might reflect moments of injustice (see Figure 2).

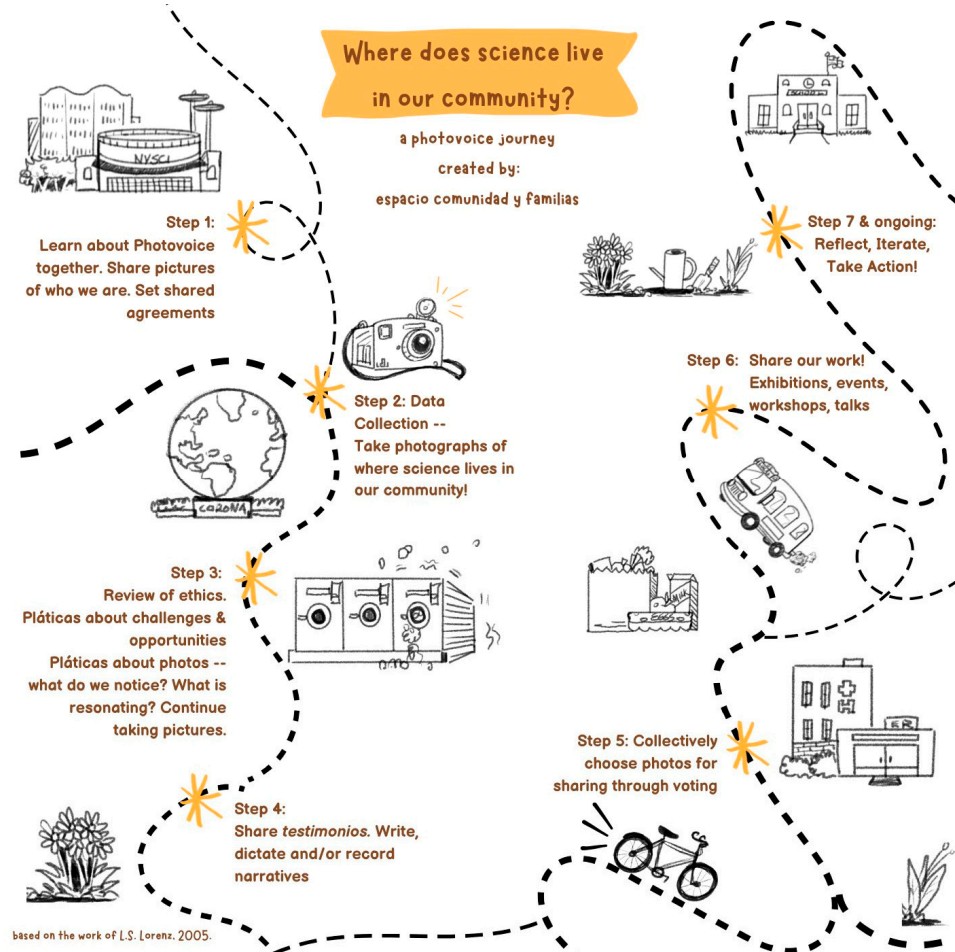

**Figure 1.** Our photovoice journey; adapted from Lorenz (2005); custom artwork by Ocelotl (2024).

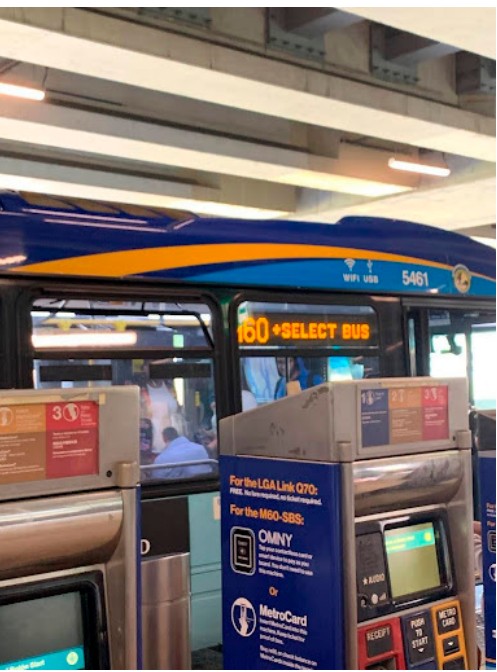

**Figure 2.** M60 select bus and OMNY/Metrocard vending machine.

Figure 2 is an illustrative example introduced by a compa to show how captions can add additional meaning to photos. The compa shared the following reflection:

There's lots of science and technology in this picture, but that's not what it's about at all. When we first moved here, I got kicked off of the M60 bus for not having a ticket. I hadn't noticed the ticket machines and didn't know what to do when there wasn't a Metro Card box (Where I came from had boxes at every door.) So I just sat down thinking, stupidly, "Wow, I guess it's free to go to La Guardia?" Pretty soon a not-very-nice cop explained to me that, no, it was not free as he checked my ID and stood over me to make sure I bought a ticket. Fast forward to my daughter's first ride on the M60. A bunch of Black kids got on and at the next stop two young white women got on. Pretty soon two cops came on board, checking for tickets. A stream of folks, all of them people of color, left out the back doors. The women didn't have tickets, but they weren't told to get off the bus. My daughter, who knew my story, turned to me and asked, "Why didn't they get kicked off?

Community researchers were encouraged to bring their children with them to the photovoice meetings, as childcare can be a prohibitive factor when engaging in participatory research. Child-centered activities were developed and provided during each session, and staff led children to a separate space, allowing the adults to fully immerse themselves in the plática sessions, engaging fully without the dual responsibilities of participation and caregiving. Creating this space was transformative as it minimized interruptions to discussions among compas and community researchers while also allowing their children to participate in parallel science-based activities. To further enhance the children's experience, successive meetings engaged in various STEM activities that related to photography or images, and we provided time before starting a session for children to share what they had engaged in the previous session. Caregivers shared that the children became very excited to attend successive sessions and were disappointed when sessions were scheduled during the school day, and they were unable to participate. In this community, engaging the whole family has proven to be highly valued in NYSCI programming and engaging in CBPR.

### 2.2. Data Collection

Data collection involved the Espacio group in creating, gathering, organizing, and discussing the role of science using photography as a tool that guided our plática conversations. Institutional Review Board (IRB) approval for the photovoice project was granted from BSC. All community researchers provided written consent for participation in the project as well as consent for use of personal photos. Photos were taken either on point-and-shoot cameras or community researcher's personal cell phones. Community researchers then sent the compas their photos via email or text message or had the option to upload their images directly from their memory sticks during the photovoice sessions.

Community researchers were invited to think about how science lives within their family and community by documenting their surroundings, focusing on themes related to where science appears in their communities. Specifically, they were encouraged to think about positive and negative experiences science has had on their personal lives. Each community researcher family was given a point-and-shoot camera; however, many families preferred using their cell phones. Each family was encouraged to take as many or as few photos as they wanted. Throughout our co-learning sessions, families were encouraged to select photographs that they believed to be most powerful and create captions describing what the image represented to them or what they intended to communicate with viewers.

### 2.3. Data Analysis

Ninety-six photographs were taken in total. Compas taped the images on wall space, and the community researchers participated in several gallery walks interacting with each photograph, caption, and theme. The photovoice sessions became a space for shared meaning-making based on these sets of images and captions, with community researchers grouping and organizing their photos into 12 themes that reflected their lived experiences. The group engaged in thematic analysis to analyze the collected data. This process followed the plática method, where community researchers were encouraged to engage in critical dialog resembling friendly and culturally appropriate ways Latine community members converse with one another. Fierros and Delgado Bernal (2016) say pláticas particularly invite communities to share ideas, knowledge, memories, and *consejos* (advice), to collectively identify and define themes in the images that are salient to the group. Braun and Clarke's (2006; 2022) reflexive thematic analysis was also closely followed at this stage. Community researchers iteratively revisited and reflected on the meanings of their photos and themes as they evolved. Compas encouraged reflection through member checking and questions.

Analysis involved multiple steps (Mooney et al. 2023). Community researchers were invited to engage in six phases of analysis (see Table 1). This served as a collaborative form of thematic coding. The compas facilitated a conversation with the community researchers, lifting themes that we noticed across community researchers' notes. Community researchers led the development of these themes by describing their photos and sharing their lived experiences, and in subsequent meetings, grouping related photographs.

**Table 1.** Thematic analysis of photovoice process (see Figure 3).

| | |
|---|---|
| Phase 1 | Becoming familiar with the data. In this step community researchers conducted a gallery walk and looked at over 100 photographs that were taped on a wall. Community researchers jot down what they noticed with post-it notes and place their comments on photos of their choice |
| Phase 2 | Generate initial codes. Community researchers grouped photos into initial categories based on image relevance to science (e.g., nature, technology, surveillance). During this process photos were moved around on a large wall and taped under the appropriate category with masking tape. |
| Phase 3 | Search for themes. Initially, the Espacio research group identified twelve categories that represented different experiences the group had with science. This process also allowed the community researchers to think about what photos were missing from the current narrative. |
| Phase 4 | Reviewing themes. Through an iterative process, the Espacio research group coded all of the images to fall under one of the twelve themes. |
| Phase 5 | Defining and reviewing themes. Once the Espacio research group agreed upon a set of twelve major themes, the group selected the top four images per theme to be included in the final analysis and photo exhibition. As discussed in the results section, the compas presented the community researchers with two conceptual models representing our collective work. This led to a more refined model with new thematic labels generated from the community researchers. |
| Phase 6 | Production. In the final stage of this process the synthesis of the results was presented in a variety of ways: community exhibit, BSC College Alumni presentation, NYSCI staff presentation. The Espacio group is continuing to work on producing different forms of dissemination that include written reports, manuscripts, and conference presentations to share the learnings with a broader audience. |

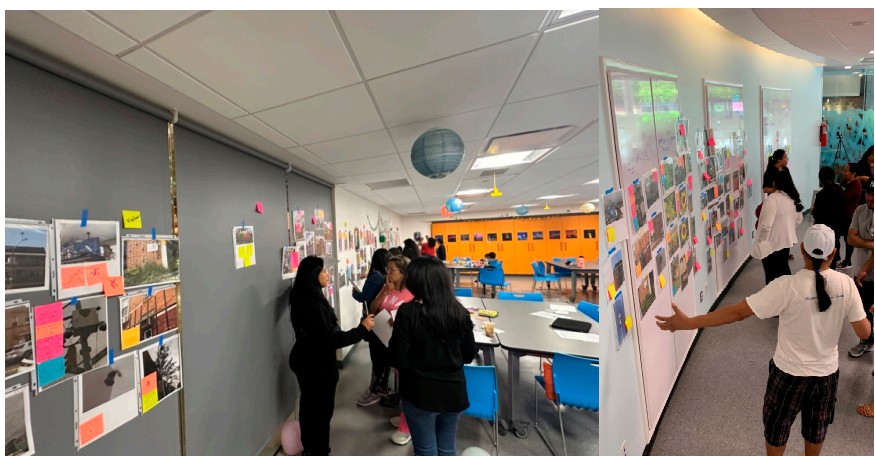

**Figure 3.** Photovoice thematic coding and gallery walk.

In keeping with the photovoice process, compas introduced the SHOWed method (Wang 1999) to the community researchers to guide the picture taking, caption writing, and analysis process by asking five questions: (1) What do you **S**ee here?; (2) What is really **H**appening here? (3) How does this relate to **O**ur lives/community?; (4) **W**hy does this condition/issue **E**xist?; (5) What can we **D**o about it? From this process, a total of 35 photos were identified as central images through a voting process that included community members and compas. Images that received majority votes were selected. Each photographer was guided to write their own caption using the SHOWed method. Similar approaches to data analysis have been observed in prior photovoice research, with theme interpretation starting from community members' perspectives, followed by whole group interpretation (Faucher and Garner 2015; Tsang 2020). This approach was especially important in ensuring that the analysis reflected and uplifted the community researchers' perspectives rather than adhering to Western research norms. Rather than prioritizing or imposing preconceived concepts and theories of the academy, the compas encouraged a deeply relational and dialectical space. The compas then provided preliminary theme organization of the project photographs (see Figures 4 and 5), which was later deconstructed and reimagined by the community researchers (see Figure 6). The multiple rounds of reflexive thematic analysis provided opportunities to challenge the biases and assumptions that the compas might inadvertently bring to the work. The group agreed on 12 categories (transportation/streets, nature and community gardens, reclaiming and reviving environmental awareness, funds of knowledge, health and future, sanitation, physical activity, technology and solar panels, surveillance, technological awareness, gentrification, artistic representations) that were further refined into three themes through collective thematic analyses during plática sessions. These categories represent the ways that science is activated through community and history (Connection to the community—*Conexión en la comunidad*; Community wellbeing—*Bienestar de la comunidad*; Resistance through knowing—*Resistencia a travez de conocimiento*). Community researchers then chose four key images per theme through sticker voting to be included in final analyses as well as in a culminating exhibition of the group's work.

This project allowed us to think more critically about the authenticity of our methods and ways to better design and implement PAR projects in the future. According to Smith et al. (2024), authentic PAR requires participants to be active decision makers in all elements of a study. Some ways this project integrated community decision making was by allowing community researchers to lead group discussions and decide if they wanted to have additional meetings. Community researchers expressed their enthusiasm about continuing this work and resulted in our cafecito series where community researchers continued to lead conversations about themes discussed during the initial project.

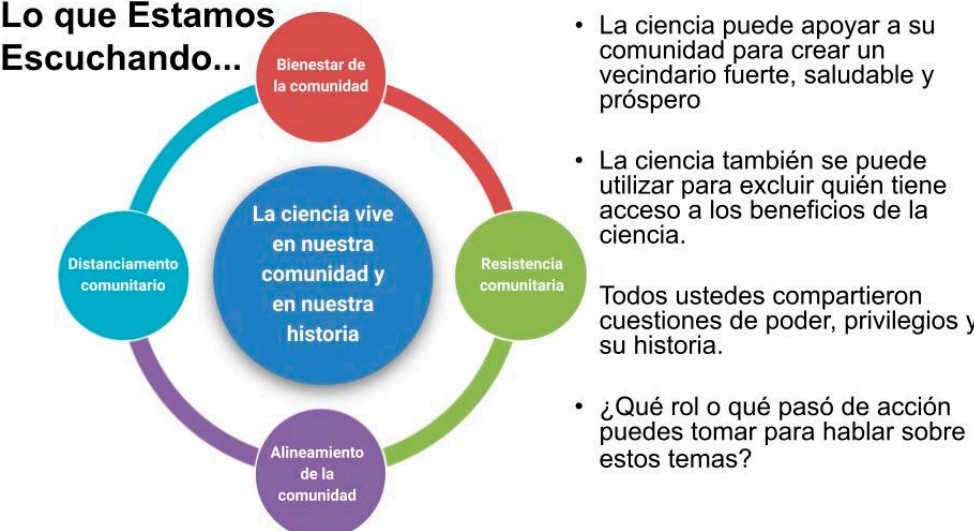

**Figure 4.** 'Compas' Initial Visualizations of Emerging Photovoice Themes (in English); colors and font size are adapted for legibility.

**Figure 5.** 'Compas' Initial Visualizations of Emerging Photovoice Themes (in Spanish); colors and font size are adapted for legibility.

The compas sustained relationship with community researchers presents opportunities to continue working towards and developing authentic PAR practices as we collaboratively ideate around ways to reclaim political power, local green spaces, and reimagine culturally situated STEM learning across a learning ecosystem. By supporting the community researchers' ability to co-lead dissemination efforts and by co-constructing future project designs, we aim to create a foundation for deeper capacity-building and meaningful long-term engagement. This model lays the groundwork for future cycles of community-engaged research and processes, in which community researchers are anticipated to take on expanded leadership roles and contribute directly to the design of new projects.

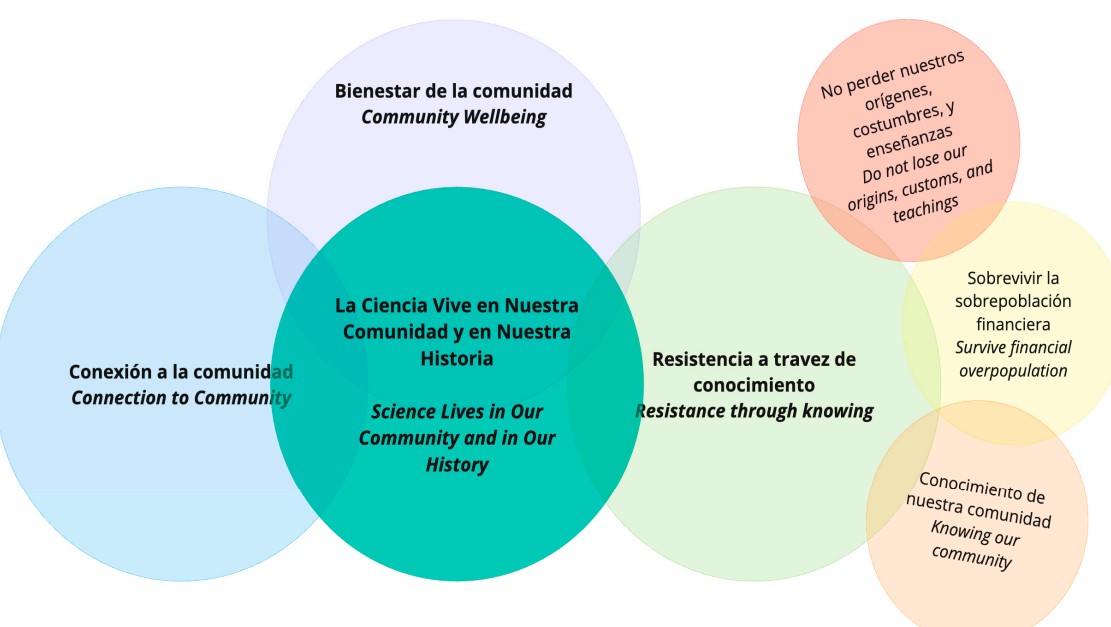

**Figure 6.** "Science through Community and History Model" Developed by Community Researchers and Compas.

*2.4. Collective Writing and Manuscript Preparation*

Community researchers and compas played an integral role in planning, drafting, and reviewing the manuscript, ensuring that it accurately reflects community researcher priorities, experiences, insights, and values that were central to this project. As active co-authors, the community researchers participated in multiple stages of manuscript development. We hosted four writing sessions for Espacio, both in person and over Zoom. One of these sessions was used to plan for a conference session, which ultimately informed the structure of this manuscript. In addition to the Espacio sessions, three additional formal meetings were held for compas to organize, remember, and write together. Quick informal check-ins between compas and community researchers over WhatsApp, Slack, Facebook, text, e-mail, Zoom, and phone calls helped to inform this work.

Community researchers provided direct feedback during conceptualization of the paper, and in early drafts. Ensuring that the community researchers had the opportunity to participate in the writing and dissemination process was critical to the integrity and participatory nature of this project.

Epistemic disobedience, plática, nepantla, and buen vivir were key conceptual frameworks identified by the compas after photovoice sessions had been completed. The compas did not come into the project with these frameworks in mind. Rather, these were identified through literature reviews and connections with other scholars as we shared and more deeply explored the Espacio research journey. For example, 2 compas were introduced to the plática method during a conference session in Spring 2024, after the project had concluded. After the session, the compas shared their excitement around learning that there was relevant and aligned literature. It felt like a happy coincidence that we began every photovoice session with "*vamos a platicar!*" or "let's chat!" This just happened to be the way we as Espacio members show up and exist in all spaces, not just research spaces. This introduction to plática re-introduced us to Anzaldúa's work, and to the work of womanist scholars like Keating, Torre, and Ford.

As the compas identified possible frameworks that aligned with Espacio's work, they were introduced to community researchers for further discussion. This iterative, dialogic, and dialectic process in which concepts were introduced, discussed, and re-

shaped to fit the community's experiences was essential to (and a great example of) the collaborative knowledge-generation process in this project. The learning was collective and often happening in real-time. In this way, both compas and community researchers informed both the content and framing of this manuscript.

## 3. Results

By asking, "where does science live in your community?" each community researcher was invited first to notice the science that was present around them. The ambiguity of the word "science" was itself an opportunity for nuance. By beginning first with images (rather than scientific vernacular or concepts), Espacio members were able to uplift the elements in their lives that are meaningful, or interesting, and maybe overlooked. In capturing and sharing these everyday images, curiosity was activated. Through plática, the community researchers began to ask each other questions, and to share their own experiences and concerns about science's impact on their *vidas cotidianas* (daily lives). Through the photovoice process and critical dialogs, several key themes emerged that reflected the community researchers' understanding of science and its role in environmental justice and community wellbeing in their urban environment. In the second and third sessions, photos taken by community researchers displayed on the wall to discuss the meanings behind their photos and sorted them into themes they identified. Compas reflected on these discussions and introduced a framework to organize them into macro and sub-themes. Through multiple rounds of thematic analysis during pláticas, Espacio began to construct a conceptual model—Science lives through community and history—that incorporates the twelve themes referenced above. This model offers examples on how science is experienced and prioritized by community researchers: (1) Connection in the community; (2) Community wellbeing; (3) Resistance through knowing (see Table 2). Where the compas saw themes fitting neatly into a 2-dimensional model (science supporting communities vs. science excluding communities), community researchers prioritized their vision of science in strengthening communities and developed a multi-dimensional model.

**Table 2.** Themes within the Science through Community and History Conceptual Model.

| Macro Themes | Sub-Themes |
|---|---|
| Connection in the Community/ Conexión en la Comunidad | Transportation and roads/ Transporte y vías |
| | Nature and community gardens/ Naturaleza y jardines comunitarios |
| Community Wellbeing/ Bienestar de la Comunidad[6] | Nature and community gardens/ Naturaleza y jardines comunitarios |
| | Reclaiming and reviving environmental awareness/ Retormar y revivir conciencia ambiental |
| | Funds of Knowledge/Fondos de conocimiento |
| | Sanitation/Sanitización |
| | Benefits of being active/ Los beneficios de estar activo |
| | Technological Awareness/Conciencia tecnológica |
| | Surveillance/Vigilancia |
| Resistance through Knowing/ Resistencia a travez de Conocimiento[8] | Do not forget our origins, customs, and teachings/ No perder nuestros orígenes, costumbres, y enseñanzas |
| | Surviving the financial overpopulation/ Sobreviviendo a la sobrepoblación financiera[7] |
| | Knowing the history of our community/ Conocimiento de la historia de nuestro comunidad[9] |

Background colors are associated with the colors and mapping in Figure 6.

The following subsections include direct statements from community researchers, which have all been translated from Spanish to English for this paper. Community researcher names were anonymized with letters to protect their privacy.

*3.1. Connection in the Community: Transportation, Streets, and Green Spaces*

Community researchers identified the critical role that transportation systems and public streetways play in both connecting and fragmenting the community. Initial conversations recognized how science and new technologies were transforming community infrastructure such as their local sidewalks, roads, and public spaces. For example, community researchers noted how public spaces have been transformed through technological advancements, such as the implementation of CitiBikes, which community researchers connected to broader discussions about technological innovation and urban planning. Their images of local streets being filled with CitiBikes throughout the community also served as a catalyst for conversations around the beneficiaries of these developments (see Figure 7). For example, through our pláticas, community researchers wondered how and why projects like CitiBikes are imposed on communities without their feedback or consent.

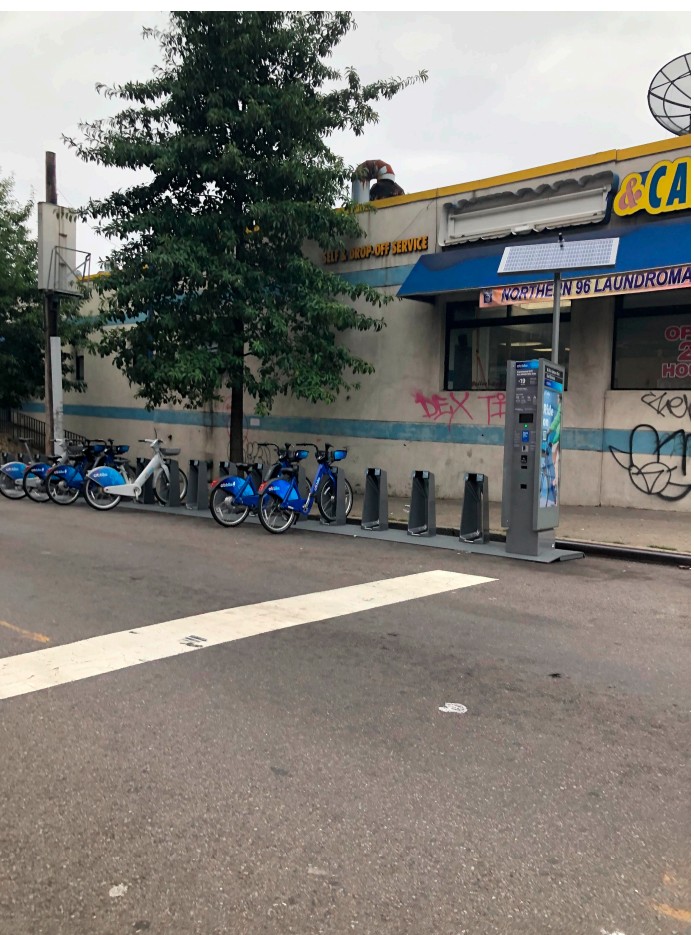

**Figure 7.** Community researcher's photograph of CitiBikes.

The developers may have imagined that this project would serve to connect people across the community; however, as we began pláticando, the community researchers identified that the introduction of CitiBikes in fact served to disrupt their communities by occupying space for street vendors, parking, or sidelining expansion of green spaces, walkways, and street cleanup. Additionally, community researchers noted barriers to CitiBike access, including the need for smart phone technology or credit cards. Community

researcher B expressed that she felt the CitiBikes would not be accessible to her or the community because they were unfamiliar and costly:

> Northern Boulevard, between, say, 103rd and 90th. I was so surprised at how many bikes I missed. I look around because I have no knowledge of those bikes. . . the electric bikes are not free for the community. So when they raise the transit fare for the community, who's going to use those bikes? My community? No. I don't know who's going to use them. And how can you invest in that? On every two or three streets, there are more bikes. And there is trash on the sidewalk, where no trash can exists. The city has disappointed us.

Another example of this fragmentation was present in the images of roadways that seemingly de-prioritize community needs such as life-saving services (see Figure 8). Community researcher *C* shared her own solution and idea for street improvements that would be most beneficial to her community: "...just like they did this for the buses, I would also like them to invent [a lane] there [for] the ambulances, the police, when there are emergencies, because in some way it also favors us, right?..."

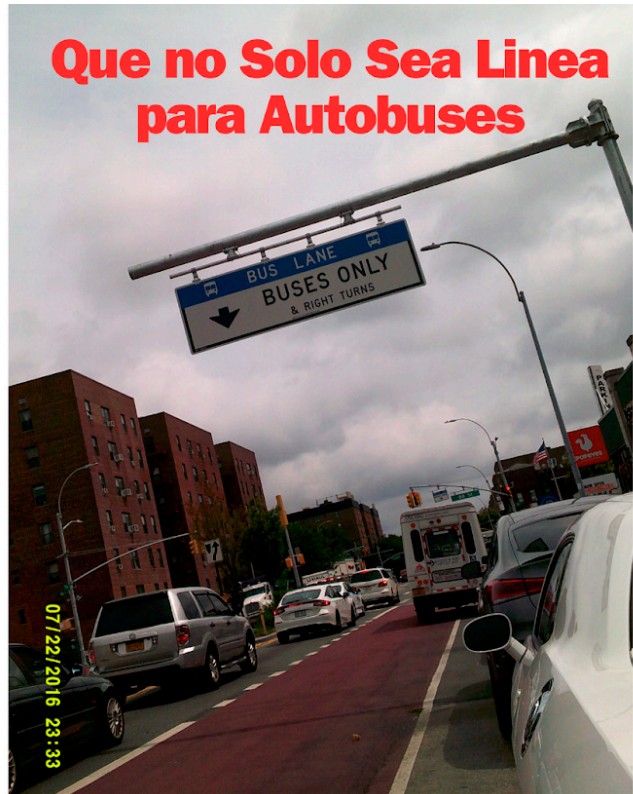

**Figure 8.** Community researcher poster: It should not only be a lane for buses[10].

In a seeming contrast, access to green spaces was highlighted as a vital asset that promoted health, community connections, and cultural perpetuity. Photos of nature were popular among the group, with a total of 21 photos depicting flowers, fruits, and vegetables. Food, in particular, represented a site of cultural expression and transmission of ancestral knowledge for Espacio (see Figure 9). The vibrant images of fresh foods in community gardens led the Espacio collective to think critically about food and green space accessibility, and the initial invitation to explore where science lives in our community led us to conversations around science education that encourages relationship and contact with the land (see Figures 9 and 10). In describing the meaning behind these images, many community researchers also identified the role that green spaces play in promoting mental

health, with one community researcher noting "The importance of being active, the physical and emotional benefits. . . the wellbeing of the community." The community researchers described how spaces like parks and gardens serve as a kind of sanctuary, particularly for families, who use these spaces to practice physical activity, to gather socially, to learn about life cycles, and to reflect.

## Que Hay en el Supermercado?

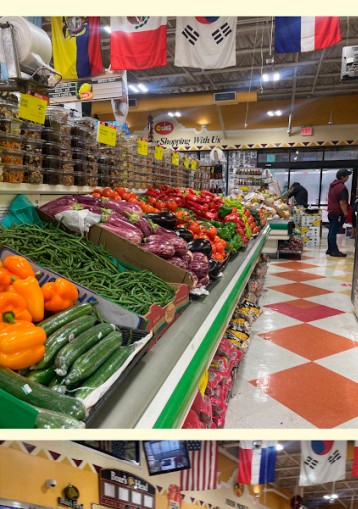
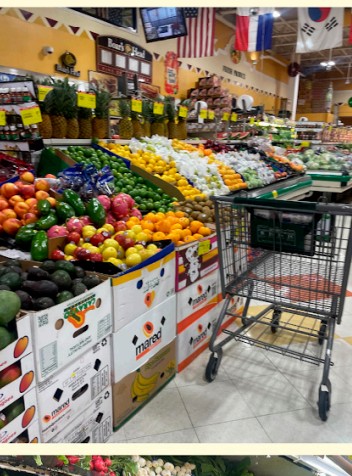
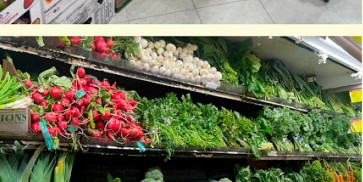

El supermercado es un lugar en donde podemos encontrar una variedad de productos de diferentes países. Es un lugar con una mezcla de culturas y esta lleno de muchos colores.

El supermercado es un buen lugar para compartir con nuestros niños, con nuestra familia buscando ingredientes para nuestras recetas que despierten los sabores de nuestras tradiciones y cultura.

La visita a los supermercados puede ser una gran oportunidad para explorar juntos el mundo motivando el amor por el aprendizaje de nuestros hijos.

**Figure 9.** Community researcher poster: What is in the supermarket?[11].

While discussing their photographs, the group emphasized the importance of maintaining and expanding access to green spaces in order to ensure that all community members could benefit from them. The group recognized that nature created opportunities for STEM learning and social connectedness for the whole community. Multiple community researchers such as *B* described her own involvement in local community gardens and argued for the necessity of activating additional green spaces for the community's benefit. At times, the group directly contrasted the value of these spaces with other forms of urban infrastructure that seemed unnecessary or inequitable.

So, those bikes are electric, right? If you have to pay, you pay not with cash, you pay with [credit] card. It has its own solar panel. And I don't have a credit card.

So, that's the truth, that's my concern. Yes. . . for my community, right? So, and the community garden, which I saw, well, posted in the photo. For me it is an emergency. It is an emergency to have those spaces. Not so much because of the food, yes, we know that food hinders a lot, but at what cost? Us. So, yes, if we had those little community gardens, just like the bicycles, on every corner, every block. Then, yes, we could harvest the basics. And also, that in connection with the land, which our children need.

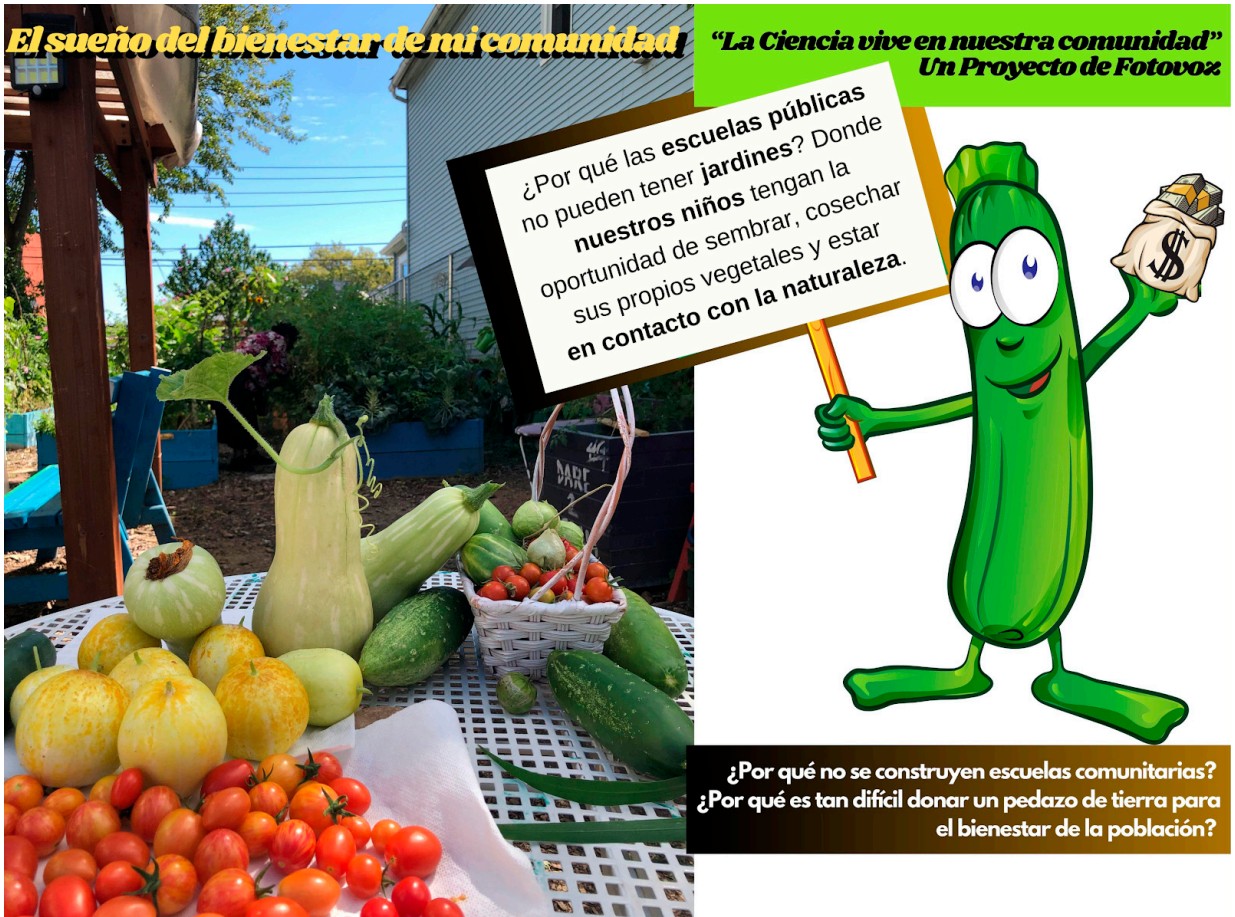

**Figure 10.** Community researcher poster: The dream of wellbeing in my community[12].

*3.2. Community Wellbeing: Funds of Knowledge, Sanitation, Technological Awareness, Reclaiming and Reviving Environmental Awareness, Benefits of Being Active, Health and Future*

Community health is inherently tied to the knowledge and practices of the women who sustain it. From the beginning of the project, and throughout its life, the community researchers called in the funds of knowledge (Gonzalez et al. 2005) that helped to shape their understanding of science. They shared the ancestral traditions of horticulture, embroidery, food, and even laundry, noting how these practices have shaped their relationships with the environment and their communities. The community researchers noted how community health is shaped by a combination of factors, including sanitation and access to clean spaces, education, and the passing down of cultural practices such as shared knowledge about health food growth and preparation.

Lived experiences became central to understanding how community members navigated contemporary community issues, particularly in relation to children's futures. For example, concerns about the lack of continuity between school curricula and the opportunities for learning offered by shared green spaces served as an integral conversation

during our pláticas. Despite the challenges they described, the community researchers' contributions resounded with resilience, with women often facilitating efforts to share knowledge and resources with one another and their families (see Figure 11). Community researchers share the benefits of having parks and green spaces for their children to play in (see Figure 12). Community researcher D says:

> Also talking about the weather, because sometimes it is very hot now. When children go to the park, they love to get wet and enjoy the trees, the shade of the trees, the plants, the park. Yes. Because children are very happy when they go to the park. All children are happy in the park.

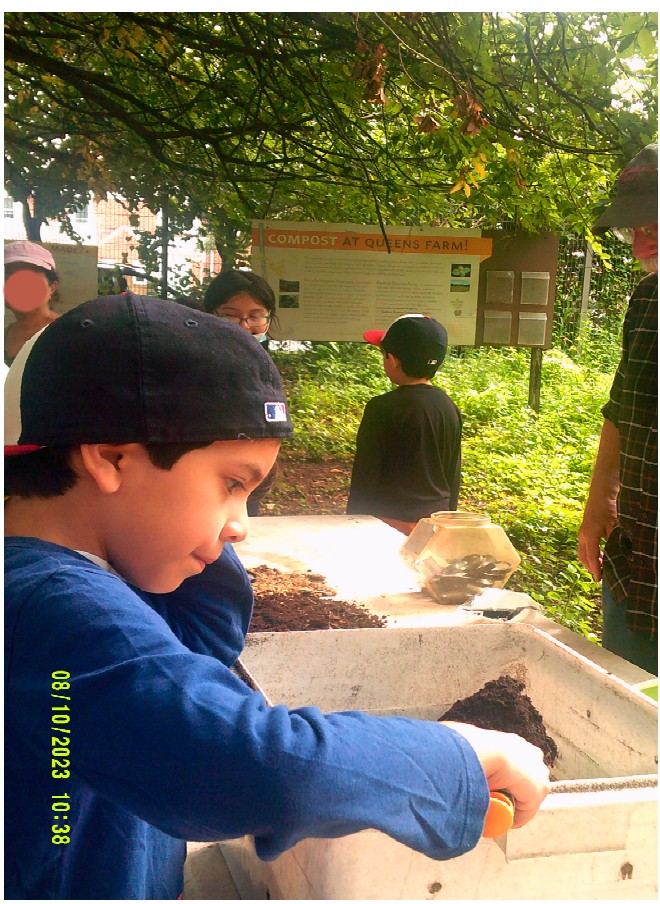

**Figure 11.** Community research photograph: How to help the planet/composting.

*3.3. Resistance Through Knowing: Surveillance, Technological Awareness, Gentrification, Artistic Representations*

During the second meeting, community researchers learned about informed consent and how to properly ask local residents and businesses for permission to be photographed. This discussion created opportunities for community researchers to reflect on their prior experiences with unethical photography and video-recording practices in their community. These initial conversations around photovoice ethics served as an entry point to critical consciousness-raising, and a collective exploration of the power dynamics at play in the local community. Some community researchers such as *A* expressed how they felt exploited and powerless when a photographer with police support took pictures of their children at a park without their permission:

> Well, it's what you say, but me, my children [go to] the park on 103rd, 104th there. A photographer, I imagine a professional one, with a camera…took pictures of all the children. Nobody said anything to them and nobody freaked out and the

park was full. Of course, they were white. White, they took pictures of all of them, well, I tell my sister, 'they took pictures of us, the *cucarachero* [bunch of cockroaches] here, let's see what reaction we get', I tell her. And nobody says anything. Yes. So he [the photographer] didn't ask permission. No, he grabbed it and stood up, I was there because I stand there. [the park] It's full. It's full all the time...The police were there and didn't say anything. I went to ask them directly...They asked them and they said they had permission, permission to take photos. I said, 'but whose permission?' The police were at both entrances, yes, the two main entrances were the police. But no one told us as parents or as the community. Well yes, the community, like that, the *cucaracha*, that's why I tell my sister, we are *cucarachas*, now yes, you see a *cucaracha*, you make it like that and it moves and he runs and comes back and itches here and it moves. They tell us that. No, but it's not right.

## "La Ciencia Vive en Nuestra Comunidad"

### COMUNICACIÓN EN SILENCIO

Las personas descansan en el parque de "Little Island" un lugar tranquilo donde puedes encontrar paz interior y también puedes ahorrar energia electrica respirando aire natural.

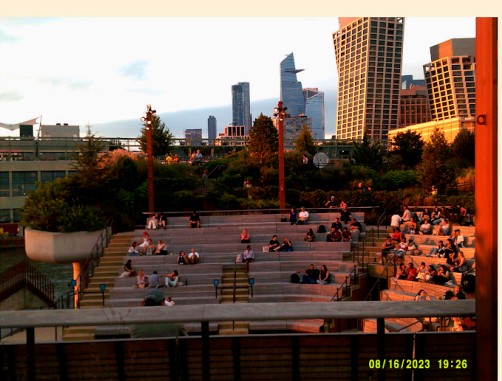

### LA IMPORTANCIA DE INSECTOS

El 80% de las plantas con flores están especializadas para la polinización por variedades de insectos entre ellos las más importantes son las abejas. La polinización es crucial, porque muchas de nuestras verduras y frutas y los cultivos que alimentan a nuestro ganado dependen de la polinización, las abejas también extraen néctar de las flores frutales por tal motivo que producen la miel para el consumo humano, ayudemos a cuidar el medio ambiente ya que las abejas están en peligro de extinción.

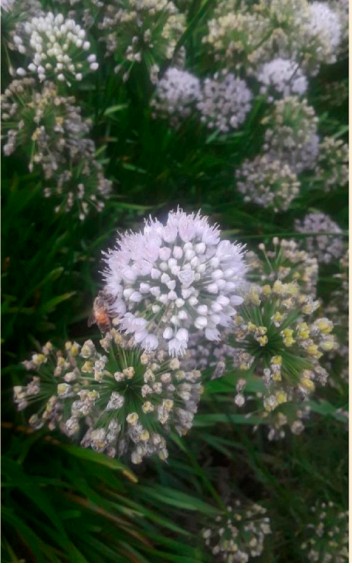

**Figure 12.** Community research poster: Science lives in our community[13].

The community member described how they felt powerless and unsafe when trying to understand why a photographer was taking pictures of a park full of children and why the police were defending them without presenting permission or reasonable explanation. While it is legal to photograph subjects in public spaces in the United States, the community

researchers felt that the ethical guidelines used in photovoice should apply to all kinds of public photography. They also alluded to how people with privilege (such as the white photographer) were not held accountable by the law. During this discussion, another community member said: "There is law, but there is no justice." These early pláticas around the shared values of consent and noticing where science lives within community invited the Espacio group to explore a more nuanced understanding of the intersectionality of power and justice as they relate to STEM.

The community researchers' reflections on technology and surveillance raised important questions about the ways that technological advances are often used to control the movements of vulnerabilized populations and the opportunities the group identified to resist these developments. In addition to discussing new bicycle-sharing terminals (described above), the group also raised concerns about other surprising developments in their community, such as the increased number of police surveillance cameras at new buildings (see Figure 13). Some community members felt police cameras provided additional security to their neighborhoods while others felt they could be used to spy on innocent residents:

> Police continue to create insecurity, well, with the cameras, no? We don't know if they are being used to spy on the criminal or the innocent. So...I see the police security cameras that should...make us feel safe as a community and this isn't the case many times. Insecurity is present, the police are watching you.—Community researcher C

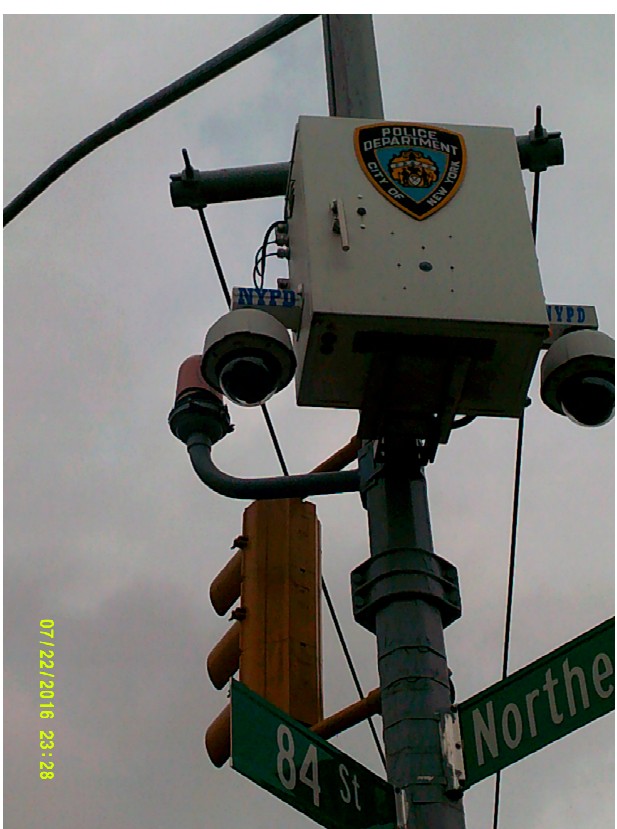

**Figure 13.** Community researcher photograph of an NYPD traffic/security camera.

Even "green" developments such as solar panels might appear to be an eco-friendly commitment, but in fact resulted in the displacement of long-time residents through rent inflation. As the group reflected on how to describe their concerns, they reflected together about how terms like "gentrification" related to their own lived experiences (see

Figure 14). For example, during our second meeting, community researcher *E* defined the term gentrification for those in the group who were not familiar with it:

> The word is "Gentrification" in English. . .Does everyone know what it means? Well, it was explained to me by my first English teacher. But what happened with the Brooklyn community, this thing about the big chains starting to arrive. They are starting to raise the prices of the rents, the rental companies. We have almost the same prices as in Manhattan practically now, and it's all because of the new constructions that are being promoted here In Willets Point. The rents are going up. Things are getting more and more expensive. They are pushing us out of the neighborhood. Now I'm struggling to get a rental. . .we also see that there are new buildings, people who have moved with a lot of money, and new surveillance appears, like private security but in reality, it is the police. We all need private security and not just because a new building is being installed, right?

While most community researchers were unfamiliar with the term *gentrification*, they immediately recognized its impact on their housing, finances, and community spaces. This was a recurring theme in many of our following conversations. After a number of conversations where the theme of gentrification emerged as a series of processes that directly impacted the community researchers' lives, the Espacio team ultimately voted to name this phenomenon *sobreviviendo la sobrepoblación financiera* [surviving financial overpopulation]. This shift from Western, academic jargon to language that more closely aligns with the lived experiences of the community researchers—specifically, the experience of survival—was a necessary intervention in our understanding of the phenomenon. Rather than focusing exclusively on the infrastructures of oppression, and specifically the economies of dispossession (Byrd et al. 2018), Espacio focuses on the infrastructures necessary for community posterity. We discussed the visible indicators of gentrification through Espacio's new lens, naming and critically analyzing the systems that negatively impact our communities, and through our relationality and heightened critical consciousness, we have been motivated to imagine how urban developments could be harnessed to support community wellbeing.

*3.4. Planning Towards Action*

PAR methods, including photovoice, provide powerful containers for community members to share stories and better understand the contemporary issues that impact their daily lives. Sustaining the momentum of our group's work was a priority, and Espacio has sought to create pathways towards future actions, including dissemination and ongoing planning through regular, informal meetings. Relationality was foundational to our work, and as such Espacio decided to further develop relationships, understanding, and next steps for action and advocacy through our Espacio meeting series—a set of cafecitos where community researchers and compas could catch up, hold dialogs about existing community concerns, discuss findings from our photovoice project, and plan actionable community projects that leveraged the findings. The cafecitos included anywhere from six to ten community researchers, compas, and community members who were not previously involved, including relatives and community leaders. To date, Espacio has hosted 4 cafecitos to further discuss surveillance, gentrification, and the reclamation of green spaces, and plan potential action steps for our group on these issues. Our most recent cafecito was hosted at a local community garden, where community researchers unpacked the local histories of this green space and considered what further activation of these spaces towards civic engagement, critical dialog, reclamation and exercising of political power, and justice might look like. When listening to community researchers discuss how to reach their local representatives to take action, one compa suggested the following:

About bringing these mayors, bringing these people… asking them, hey, I want this change in my community, right? I see that you have a lot of these bikes, but where are the gardens? So, that's an action, right? That we want these people to take. So, think about these actions.

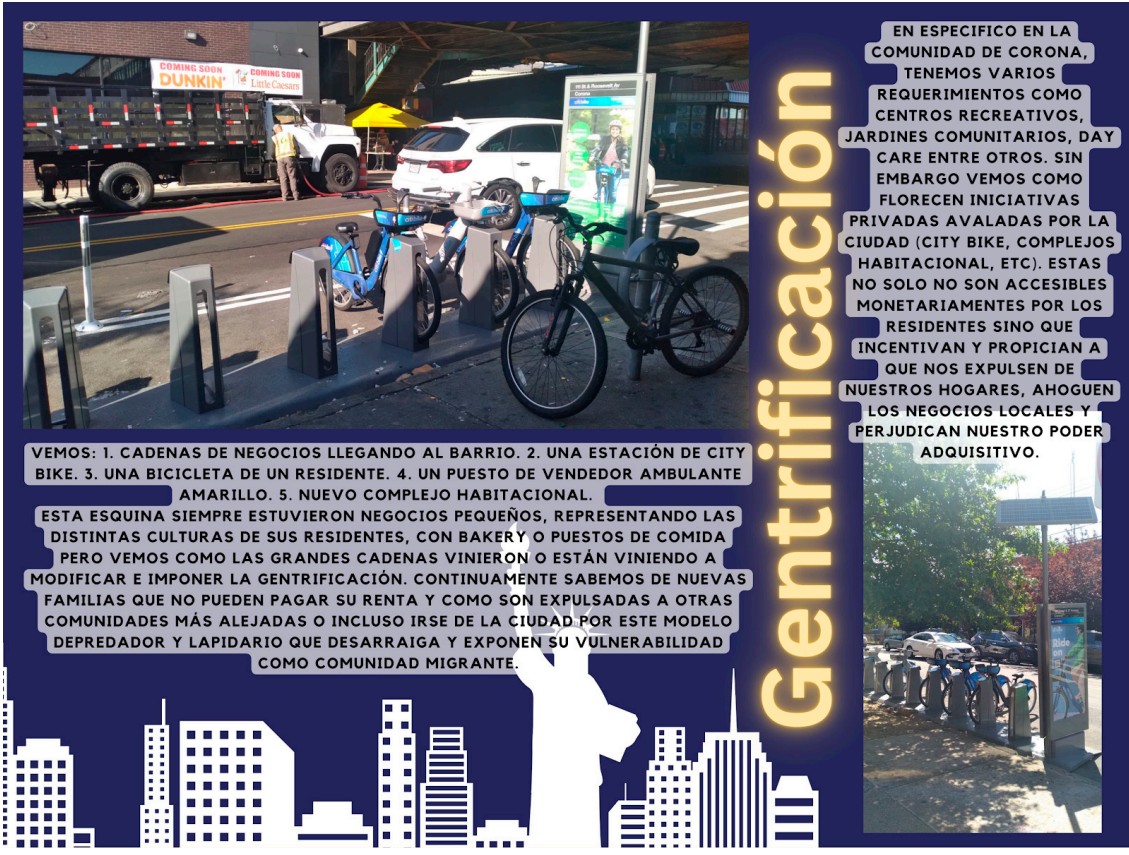

**Figure 14.** Community Researcher poster: Gentrification[14].

## 4. Discussion

Settler ways of knowing generally center a positivist paradigm that advances the notion that ways of knowing are based on a single truth and reality. Indigenous peoples, on the other hand, are more likely to recognize that there is no single reality. Consequently, Indigenous peoples throughout the world have developed complex ontologies of being closely linked with place that mirror contemporary concepts of quantum physics that view the universe as constant flow that is both infinite and localized… (Clarke and Yellow Bird 2021, p. 81)

Western science education often disregards experiential, spiritual, embodied and cultural ways of knowing, favoring institutionalized knowledge formed outside of communities. This project seeks to integrate these overlooked ways of knowing, guided by the DataCenter's (2023) Research for Justice framework, which values cultural and spiritual traditions, lived experiences, and community wisdom as essential sources in the knowledge production process. Our findings highlight the importance of epistemic disobedience in developing a more nuanced understanding and approach to science. We began with a single question: where does science live in your community; the subsequent conversations evolved unpredictably according to community members' lived experiences. Asking community members to identify where science exists in their everyday lives is asking them to notice their lives, their relationships, their struggles, and their hopes and aspirations. Noticing our everyday lives through the photovoice method activated our radical imaginations,

allowing us to see science as a connective force that can enhance our relationships with our families, communities, and ecosystems. These observations of science confirmed for us the important role of community voice in decisions around urban planning and land use and the central role of green spaces in affirming community expertise and agency and fostering community wellbeing. For our community researchers, green spaces represented opportunities to (1) build and maintain community; (2) to preserve and transmit [pre-colonial] onto-epistemologies; and (3) engage in multigenerational scientific inquiry.

These threads connecting the group's ongoing discussions align closely with principles of buen vivir. Economies of buen vivir exist as an alternative to the capitalist aims of profit maximization, wealth accumulation, and industrialization (Alcoreza 2013). This cosmocentric framework is relevant to the project's focus on investment in green spaces for community wellbeing, and the group's discussions about how environmental stewardship and urban planning might prioritize community needs and values over those of wealthier/powerful stakeholders. The group identified green spaces in Corona-Flushing as sacred and integral to their cultural identities and aspirations, and as a site for ongoing multigenerational learning to strengthen and sustain cultural traditions and ways of knowing.

Photovoice as a method for raising critical consciousness among community researchers has provided a new lens for community members to examine the ways science appears in their daily lives. Combined with feminist theory, photovoice was an accessible and interesting entry point for community members to share their opinions on community assets and barriers, analyze the often unequal power dynamics between themselves and local decision makers, and how these elements could lead to both beneficial and harmful social, environmental, and health outcomes. Prior studies have justified using photovoice as an appealing and flexible nontraditional research method that allows participants to express their knowledge, thoughts, and ideas in visual and creative ways not typically offered by traditional interviews and focus groups (Nykiforuk et al. 2011; Lantos et al. 2021). Photos taken by community researchers can also allow institutional researchers to visually communicate how specific community elements such as surveillance, gentrification, and nature can be addressed by local decision makers. The photovoice method aligns with our decolonial approach to acknowledge how traditional Western research framings could implicitly limit what community researchers share and allowed them to take the lead when interpreting where science exists in their community (Flores Carmona et al. 2018). This project created a landscape where community researchers could share the complexities and nuances of their lived experiences. Together, Espacio team members explored the damage that our community has been subjected to, though the compas and community researchers resisted a deficit-based research project. Because Espacio was able to capture moments of beauty, resistance, and resilience, we were able to move towards and engage in desire-based research (Tuck 2009), which enabled us to imagine the futures and wellbeing of our community and the ways that science might help us arrive to those futures.

The regular sessions and pláticas emphasized the interconnected nature of ecological systems, and both the challenges and opportunities that community researchers confronted in practicing agency and advocacy in these spaces. Building on Bronfenbrenner's (1979) socio-ecological model, and Fish and Syed's (2018) Indigenist Ecological Systems Model, Espacio has begun to explore the various sites where the reclamation of political power through collective action is possible (see Figure 15). The group explored how history, culture, and power were interwoven in ways that affected all levels of the community ecosystem. Through plática, Espacio members have been able to identify opportunities to exercise not only personal agency, but collective action and political power as well. For example, this research project has resulted in a number of dissemination opportunities,

including exhibitions and workshops at BSC and NYSCI. These learning opportunities have engaged the broader community and have influenced institutional priorities, including NYSCI's future research directions, and family engagement programming. Most recently, conversations around green spaces have converged with institutional, community, and city agency interests to provide science learning opportunities through a newly established hydroponic exhibit. Students from the local Pre-K center were recently invited to harvest plants from this exhibit. Similarly, leadership and researchers at BSC have been more intentionally considering how family voices can be centered in educational decision-making processes.

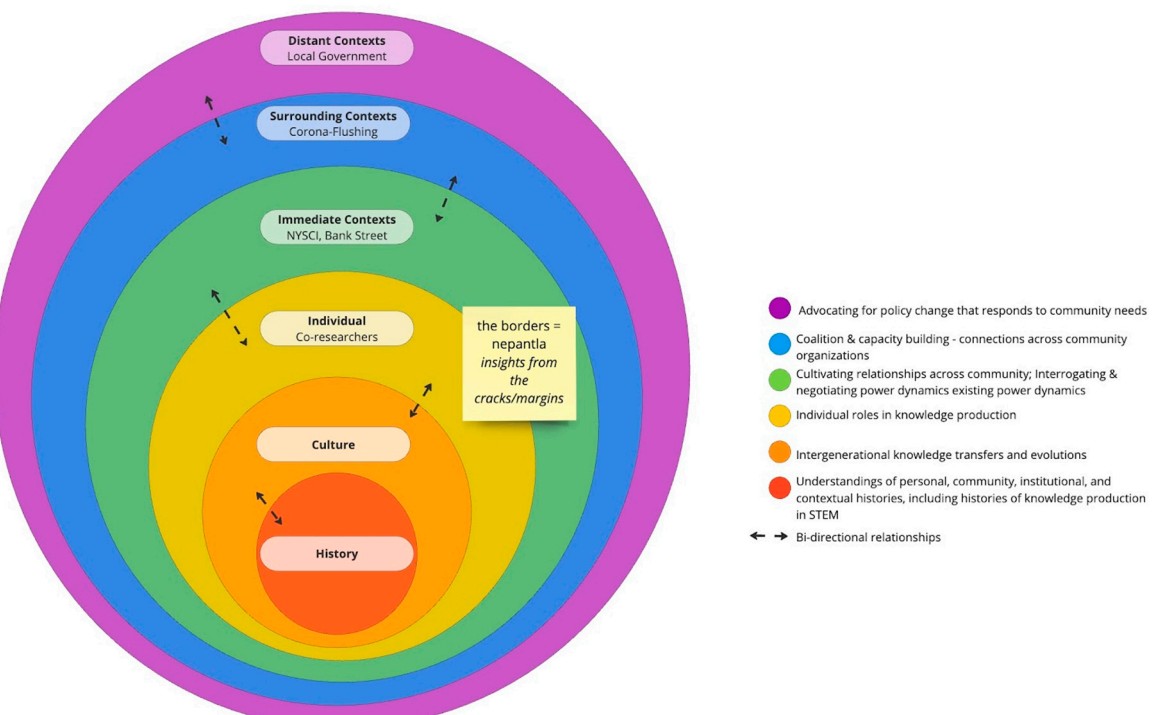

**Figure 15.** Sites of political power, collective action and agency. Adapted from Fish and Syed (2018) Indigenist Ecological Systems Model.

Through this project, Espacio teammates identified how plática as a practice in epistemic disobedience allowed us to be. Being able to exist in our complexities and imperfections allowed us to use science as a lens to identify and reconnect with Latin American Indigenous worldviews such as buen vivir. These images and stories also allowed us to redefine our connections to time and space. During our pláticas, we regularly remembered our ancestors. We grieved together. Laughed together. Experienced joy when our children shared their science learning. We ate amazing homemade tamales and *papalo* (summer cilantro) from a teammate's home garden. We thought of the relatives who have left us, and those who are yet to arrive. This project allowed us to see that all of these experiences are part of science learning. How can we talk about science without talking about a pandemic that took so many of our loved ones? How can we talk about science and not share our *remedios caseros* (home remedies) with one another? How can we talk about science without naming the spirit work? The images we took, the stories we told, and the connections we made confirmed that everything is connected. And that is why our stories and experiences are so necessary. They give us a fuller, richer, more complicated understanding of what science even is.

The members of Espacio believe that the cosmogonies that emphasize harmony between humans and nature should be incorporated and prioritized in science education.

Developing science education and research models that transmit ancestral wisdom across generations and prioritize environmental justice and sustainability are necessarily antithetical to the Western and colonial imperatives of modernity. Delinking from colonial knowledge systems creates real pathways for decolonization.

Ultimately, as Tuck and Yang (2012) warn us, decolonial desires alone can result in a perpetuation of settler-colonial logics. Decoloniality that is oriented towards justice must prioritize the original stewards of the land. As New Yorkers and residents of Flushing-Corona, we acknowledge the Canarsie, Matinecock, Lekawe, and Munsee Lenape as the original stewards of this land. In writing this paper, the Espacio team acknowledges the importance of reflecting on our roots and histories when engaging in decolonial work, especially in the study of science. Several of us are part of diasporic Indigenous communities, primarily from Latin America, and all of us are impacted by coloniality. The Espacio collective hopes that through community-engaged methods, like photovoice and plática, we (the greater population) can return to ourselves, to pluralistic and unique ways of knowing and being, and to each other. This project is an extension of the compas' commitment to cultural and community responsiveness and is encouraging compas to identify sustainable strategies and opportunities to engage with local communities, and specifically local Indigenous communities, to further guide and inform this work. We recognize that Indigenous identities, rights, and responsibilities are deeply and significantly tied to ancestral lands. With this in mind, we approach this work with humility and a commitment to accountability and solidarity with Indigenous calls for decolonization. Espacio's process of delinking from Western ideologies that dominate scientific inquiry through this photovoice project represents a step in a larger decolonial context, in which we aim to trace the threads of our interconnected histories in order to better relate to one another and the world around us (Krawec 2022).

NYSCI represented a site of nepantla by offering space for exploring new understandings of science and opportunities for transformation. Within this space, group members identified opportunities for community-driven change and guided the project towards spaces of healing and growth.

This project also highlights how CBPR projects can work to address power imbalances in the research process and counteract dominant narratives of science. The group's use of photovoice as a CBPR method positioned every member of the group as a producer of knowledge. By democratizing the research process in this project, the community researchers were provided with the time-space to identify the systems that have stripped them of power and voice. In addition, the photographs that community researchers took provided an evocative and necessary entry point into conversations about the intersections of science and justice. Reckoning with perceptions of science outside of traditional educational spaces, and without the guidance of scientific "experts", in itself served as a powerful reminder and realization for our community researchers: science is not neatly confined to experts, classrooms, or laboratories. Through their photographs, the community researchers began to recognize how their relationships to science could serve as a conduit for power through transmission of cultural identity, knowledge, and practices. This approach revealed unexpected directions for the project, affirming the legitimacy of community-driven knowledge production, and emphasizing the need for inclusive research practices that prioritize the voices of those most affected by policy and decision-making.

### 4.1. Recommendations

The project revealed opportunities and challenges of engaging community members in CBPR within institutional contexts, particularly in partnership with museums and organizations of higher education that members of the compas work within. The Espacio

team learned that while these institutions have the potential to be powerful allies in community-driven work, it is important to actively and critically examine each institution's roles in disrupting or maintaining the status quo. The compas participated in regular collective and individual reflexive practice and dialog to ensure that the space we cultivated prioritized the insights, perspectives, and lived experiences of our community researchers, and deprioritized traditional, academic, and Western narratives. The lessons learned through this project provide recommendations for researchers and educators who wish to center community voices using participatory methods (see Table 3). The project findings also have broader implications for practice and policy, namely that the voices of community members must be integrated into decision-making processes at multiple levels to ensure that urban development initiatives actually serve the needs of local populations, rather than exacerbating or perpetuating existing exclusionary practices and inequities. Here, we offer multi-level, system-specific recommendations (see Table 3).

**Table 3.** Recommendations for researchers, educators, educational institutions, and policymakers.

| | For Researchers and Educators | For Educational Institutions | For Policymakers |
|---|---|---|---|
| **Relationality** | Trust building to honor and amplify local knowledge | Create spaces for power-sharing and knowledge co-creation | Support community-driven development and policies |
| **Practice** | Avoid imposing preconceived expectations/existing understandings | Tailor strategies within the ecological systems | Promote equity in STEM education |
| **Reflexivity** | Support and build capacity reflexive practice | Engage in ongoing institutional reflexivity | Incorporate community input into policy development |

### 4.1.1. Relationality

The role of relationship building was central to this project. In order to make sense of STEM's role in social and environmental justice, it is necessary to expand our understanding of what STEM is and could be. Embracing a pluralistic approach to STEM knowledge production not only honors diverse onto-epistemologies, but it can provide us with innovative ways to navigate some of the contemporary issues we are confronting today, such as environmental injustice. By approaching knowledge through the experiences, senses, and intuitions of local community members (Cordero 1995), this project aimed to reimagine knowledge production processes. Researchers and educators should strive to create research designs and learning opportunities that allow for the organic emergence of community-driven directions and commitments. Arts-based, community-engaged research methods like photovoice can enable participants to visually document and discuss their environment, which often leads to powerful storytelling. Elevating community voice in these ways deprioritizes knowledge production strategies that are grounded in colonial and white supremacist logics of domination. For example, prioritizing artistic expression in research disrupts "worship of the written word" (Okun 1999, p. 4) by affirming intuition and embodied ways of knowing. Embracing these different ways of knowing means challenging the knowledge creation and assessment processes and recognizing that lived experiences and local onto-epistemologies will not neatly fit into Western paradigms (Wilson 2008). Engaging in pláticas has allowed us to honor and validate experiential and embodied knowledge, creating nurturing spaces where community members feel empowered to share their personal stories and reflections. The space we have co-created has

become a space of healing and reciprocal care. Most recently, we held a writing circle where we invited community researchers to review and discuss manuscript feedback together. This session naturally evolved into a meaningful space for mutual witnessing, support, information sharing, and action planning. Holding space for one another in this way honors the emotional experiences that are informed by real, ongoing systemic inequities.

At the institutional level, it is necessary to have a commitment to cultivating spaces that address existing power structures and dynamics, creating space for tensions, contradictions, challenges, and opportunities to coexist. The concept of nepantla provides a powerful metaphor/framework for understanding why and how institutions can facilitate critical dialog and transformative action. Institutions could leverage methods like photovoice to encourage epistemic diversity and friction. Further, could incorporate focus groups or community mapping to gather broader perspectives and build more holistic understandings of the community's needs. Institutions could then use this feedback to develop programming that aligns with community-defined priorities (e.g., advocating for more opportunities to learn with nature at school as a STEM priority in this project).

Policies should support community-informed urban development projects that are driven by the needs and aspirations of local communities, rather than those that prioritize the interests of wealthier or more powerful stakeholders. Supporting these projects can uncover powerful insights. For example, unearthing concepts like buen vivir can offer policymakers a valuable lens for developing policies that promote justice and posterity. To ensure that community voice is actually heard, policymakers must establish pathways for integrating community input into policy processes. This might include funding incentives for organizations that demonstrate robust, intentional community-based approaches. Collaboration with community-based organizations that actively incorporate participatory research methods could serve to inform policy decisions. For example, Espacio's community researchers' insights on CitBike stations and the disruption of public spaces illustrates the gaps between actual versus perceived benefits of development projects.

### 4.1.2. Practice

It is crucial for researchers and educators to avoid imposing expectations, frameworks, and even language that do not align with community members' vida cotidiano. Institutional researchers are encouraged to draw on insights from community-engaged research to propose language, frameworks, and concepts for collective discussion and further development. By leveraging academic privilege to contribute theories as part of a shared meaning-making process, this approach subverts the imposing, extractive, and often imperialist tendencies of traditional academic research. By embracing the complexity (as well as the challenges and contradictions) of community-driven research, researchers, educators, and any professionals working directly with community members can arrive at more nuanced understandings and surprising solutions. This allows us to develop language that is oriented towards justice and that is of relevance and utility to local communities. It is necessary for researchers and educators to undertake this work with a sense of responsibility and accountability in order to disrupt the foundational colonizing practices and impacts across disciplines (Jiménez Estrada 2005). Additionally, researchers and educators should be committed to developing sustainable structures and processes to engage community members. Equipping participants with training and skills in arts-based research methods like photovoice can position them as leaders at the institutional level. By building these skills, community members gain confidence and the ability to share their perspectives on complex issues both at the institution and beyond.

While institutions contend with their equity and engagement work, it is crucial that the staff at these organizations recognize their role within a larger ecosystem. Equity

work should be tailored to address specific needs at different ecological levels (Ramanadhan et al. 2024). For example, in the case of informal STEM education, science centers and research universities can strategize around the needs of museum educators, school partners, teachers-in-training, and local communities. Approaching equity and justice across multiple levels ensure that the strategies are relevant and effective across various contexts. Additionally, forming lasting partnerships between institutions and community members should be prioritized. Institutions could develop pathways to involve community researchers in project governance or advisory roles to ensure institutional accountability. This would position community members as decision-makers who ensure institutional efforts align with community needs, goals, and aspirations. In the case of Espacio, the emergence of green spaces as a vital community asset that requires community protection is an example of a community need that both institutions could support. Arts-based, community engaged research methods like photovoice can help communities document green spaces and the cultural, environmental, recreational, health values that they hold. Both institutions have brought the community researcher posters to workshops, forums, and conferences, amplifying their stories. We hope that continued sharing of this work will help us to facilitate conversations with policymakers and community organizations with shared interests.

Educational policies that promote equity in STEM education can have meaningful outcomes for communities across various sites of the systems model. By supporting initiatives that engage historically vulnerabilized communities, policymakers are investing in social justice through community wellbeing; funding and resources for community-driven STEM learning projects promote the development of culturally responsive and sustaining initiatives. Promoting STEM education that is responsive to community interests could empower residents to advocate for their needs more effectively. For instance, environmental science education focused on local green spaces can strengthen community members' capacity to engage in development conversations and even propose their own solutions. Policymakers could encourage schools and community organizations to integrate community-responsive STEM education initiatives with the goals of (1) building local expertise and (2) fostering community leadership towards environmental justice. A major takeaway of this project is that the arts should also be prioritized. Arts-based methods also proved to be vital for science learning and knowledge production for Espacio. The arts can create space for a wider range of identities and ways of knowing outside of Western norms/expectations (including, but not limited to Latine groups), making arts-based learning approached more inclusive and equitable in intersectional ways for larger segments of the community. Photography allowed our participants to articulate the unique ways that science affects their lives. Policymakers could support programs that encourage artistic expression alongside STEM, recognizing that creative disciplines are in fact deeply related to STEM learning. Prioritizing both STEM and the arts encourages an inclusive, interdisciplinary model for engagement.

### 4.1.3. Reflexivity

Researchers and educators involved in community-engaged research must commit to and create opportunities for self-reflection. Research should be conducted in a way that challenges existing power structures and supports the diversity of voices represented in the community by weaving together seemingly disparate ways of knowing and being. This commitment to reflexivity and community *voice* ensures that institutional practices are not perpetuating oppressive and exclusionary practices.

Reflexivity cannot exist exclusively in a research bubble. There must be an institutional commitment to ongoing reflexive practice to ensure that efforts around equity and

inclusivity do not stop at performative representation. Reflexivity at this level involves continual assessment of practice across the institution.

This work truly becomes transformative when policymakers prioritize and include community input in the development of policies that directly impact the community, such as those related to urban planning, environmental justice, education, and public health. A possible strategy would be to create formal mechanisms for community feedback. Pressure-testing policies through community feedback can be an effective way of ensuring that these policies are relevant to community needs and aspirations. Clear follow-up mechanisms and feedback loops that allow community members to see how their insights have influenced decision making can reinforce this commitment to community engagement and accountability.

*4.2. Limitations*

While this project provided valuable insights into community perspectives on science and its connection to community concerns around infrastructure, surveillance, gentrification, and community wellbeing, the small number of community researchers involved is a notable limitation. With a small group of community researchers, the findings may not fully represent the broader community's views. This limited sample size restricts generalizability.

The qualitative methods employed, including plática and photovoice, allowed for deep exploration of key issues within the community. These methods enabled participants to share unique experiences, opinions, concerns, desires, and aspirations. Despite the limited size of the group, the data generated through photographs and discussions was rich and connected to larger community issues. The small sample size provided an intimate space for meaningful engagement. Future studies could involve larger and more diverse groups of participants to broaden the scope of community perspectives. Having a larger sample size might capture a wider range of community concerns across different segments of the population.

Managing power dynamics was an essential focus in this project, as PAR can still retain and perpetuate structural inequalities between professional researchers and community researchers, despite efforts to involve community members in more equitable ways. For example, while our group regularly met to share photos, create captions, and develop themes, the compas decided how meetings were structured and initiated the data analysis process without community researcher input. Ensuring that community members are active in all aspects of the project generates trust in the research process and enhances their ability to engage in storytelling and dialog about topics that resonate with them (Gabrielsson et al. 2022). Effective PAR projects position community members as decision-makers who shape how their ideas are represented and what should be included or excluded in our work (Cornish et al. 2023).

## 5. Conclusions

*¿Dónde vive la ciencia en su comunidad*? demonstrates the transformative potential of community-based participatory research methods to reclaim and reimagine urban green spaces. The photovoice method in particular provides a powerful container for critical dialog and consciousness-raising. Inviting community researchers to document their daily experiences, and their own interpretations of what science means to them was a powerful practice of epistemic disobedience. The process of complicating and democratizing knowledge creation allowed us to envision how local insights can inform urban sustainability, planning, and ultimately environmental and social justice.

Adopting a womanist lens, the project highlighted the role that women play in cultural preservation, knowledge transmission, and advocacy. The women of the Espacio collective

were positioned as community leaders with first-hand knowledge of complex issues like gentrification. Accounts of their lived experiences allowed the whole group to denote the structural, systematic, and interconnected nature of coloniality. Their experiences emphasized that discussions around community wellbeing cannot be separated from conversations around mental and public health, rent hikes, gentrification, food justice, and access to green spaces.

Through processes grounded in educación popular feminista, Espacio created opportunities for collective and reciprocal learning within the group. Looking forward, Espacio recognizes that community research is ongoing. We hope to engage the community more broadly and intentionally, and we hope to develop research projects that continue to prioritize community connection and wellbeing. Through participatory research and collective action, we plan to continue to cultivate spaces where *nepantla*—a state of being between worlds or paradigms—is possible, enriching our learning *and* strengthening community capacity for advocacy, relationality, action-planning, and collective action.

**Funding:** This research was funded by the Simons Foundation, project title "Corona STEM Partnership".

**Institutional Review Board Statement:** The study was conducted in accordance with the Declaration of Helsinki and approved by the Institutional Research and Review Board of Bank Street Graduate School of Education IRRB #060223, approved on 6/2/2023.

**Informed Consent Statement:** Informed consent was obtained from all participants involved in the study.

**Data Availability Statement:** The raw data supporting the conclusions of this article will be made available by the authors on request.

**Acknowledgments:** We are grateful for the cultural stewards, protectors, kin, and ancestors (including those recent ones) who have inspired this work. We acknowledge that this work is an echo of past generations and global movements for justice. We extend gratitude to our compa, T, without whom this work could not have flourished. We are grateful to T's family for sharing her with us, and we dedicate this work to the memory and legacy of C.G. We would also like to extend gratitude to the community researchers who contributed to our original investigation: M.C., M.J., and A.V., and would like to thank NYSCI compas J.H. and J.M. for planning and leading activities with community co-researchers' children during photovoice sessions, D.M. for helping us plan a fotovoz panel at a BSC Alumni event, and L.G-B. for support in organizing an exhibition of our group's work. Finally, we would like to thank the reviewers of this manuscript for their generosity in sharing time, energy, questions, and resources that have contributed to our continued growth and learning.

**Conflicts of Interest:** The authors declare no conflicts of interest. The funders had no role in the design of the study; in the collection, analyses, or interpretation of data; in the writing of the manuscript; or in the decision to publish the results.

## Appendix A

Espacio: Comunidad y Familias Author List

The author list is in alphabetical order by author last name. Relevant affiliations are denoted below.

Franklin Aucapina[1], diana ballesteros[1], Monica Bunay[3], Emma Confesor[3], Thania Gómez-Martínez[2], Rafaela Heredia[3], Sarah Ketani[1], Suzy Letourneau[1], Flaviana Linares[3], Cristina Medellin-Paz[2], Crispina Morales[3], Mark Nagasawa[2], Abundio Sacramento[3], Elizabeth Toledo[3], Xiaohan Zhu[2]

1. New York Hall of Science
2. Bank Street College of Education
3. Espacio Community Researcher

**Author Contributions:**

Co-authors participated in collaborative writing and synthesizing sessions during which they conducted reflexive thematic analysis of project data and existing literature on the topics of community-engaged research, environmental justice and urban planning, and epistemic disobedience. Individual contributions consisted of the following: Investigation: F.A., D.B., M.B., E.C., T.G.-M., R.H., S.K., F.L., S.L., C.M., C.M.-P., M.N., A.S., E.T. and X.Z; Conceptualization: F.A., D.B., M.B., E.C., T.G.-M., R.H., S.K., F.L., S.L., C.M., C.M.-P., M.N., A.S., E.T., and X.Z.; Resources/Data Curation: F.A., D.B., C.M.-P. and X.Z.; Writing—original draft: F.A. and D.B.; Writing—review and editing: F.A., D.B., S.K., S.L., C.M.-P. and M.N.; Supervision: S.L., C.M.-P. and M.N.; Project Administration: F.A. and D.B. All authors have read and agreed to the published version of the manuscript.

## Notes

1    Pedagogies of invitation are non-oppositional, emphasizing authenticity, intellectual humility, flexibility, and an open-minded attitude (Keating 2013; Ford 2023). This approach shaped the project space Espacio members entered, encouraging them to engage imperfectly, embrace incomplete and divergent knowledge, and prioritize collective and reciprocal growth. Here, we use this approach to engage our readers as members of our extended Espacio.

2    A gender-neutral term used to describe people of Latin American descent.

3    Recognizing that traditional terms like "vulnerable" can indicate a deficit-based approach, Espacio embraces the alternate language suggested by Garrett and Altman (2024) that acknowledges systemic and structural factors that create and maintain *vulnerability* and *marginalization.*

4    Compa is an abbreviation for compadre in Spanish that has been reserved to name godfathers or friends. The word has normally been used among males, but here Espacio embraces the queering of the language, an act of anti-colonial resistance. For us, to be a compa is to be a compañero/a (companion), compatriota (compatriot), colega (colleague), and friend.

5    Throughout this article, the teams at BSC and NYSCI will be referred to as compas, the project participants are referred to as community researchers, and our collective group (including both community researchers and compas) will be referred to as Espacio.

6    Vigilancia/Surveillance and Conciencia Tecnológica/Technological Awareness were two subthemes that overlapped across the macro-themes: Resistancia a travez de conocimiento/Resistance through knowing and Bienestar de la communidad/Community wellbeing

7    Code community researchers created to represent their experiences with gentrification. See results Section 3.3 for more information on how community researchers arrived to this code and language.

8    See footnote 6

9    Knowing the history of our community branched into an additional subtheme: "Representaciones artisticas/Artistic representations

10    Figure 8 caption translation: Drivers should be aware, everyone wants to arrive on time, but this lane is for the bus. Maybe a relative of yours is on that bus and they also want to arrive on time. Do not block this lane as it can be used by the police, firefighters, or ambluances. It might happen that a relative needs first aid or any kind of help. Be aware and use the appropriate lanes.

11    Figure 9 caption translation: The supermarket is a place where we can find a variety of products from different countries. It is a place with a mix of cultures and is full of so much color. The supermarket is a good place to share with our children, with our families while we look for ingredients for our recipes that awaken the flavors of our traditions and cultures. Visits to the supermarket can be a great opportunity to explore the world together, motivating a love of learning in our children.

12    Figure 10 caption translation: Why can't **public schools** have **gardens**? Where **our children** can have the chance to sow and harvest their own vegetables and **be in touch with nature**. Why are community schools not being built? Why is it so difficult to donate a piece of land for the population's wellbeing?

13    Figure 12 caption translation: Communication in Silence—People relax in the park "Little Island," a tranquil place where you can find interior peace and you can also save energy by breathing in natural air. The importance of insects—80% of flowering plants are specialized for pollination, the most important of which are bees. Pollination is crucial because so many of our vegetables and fruit and the crops that feed our livestock depend on pollination, bees also extract nectar from flowers so that they can produce honey which humans consume. Let's help to care for the environment now that bees are in danger of extinction.

14    Figure 14 caption translation: (on the left) Gentrification—Specifically in the community of Corona, we have various needs like recreation centers, community gardens, day care centers, among others. However, we see how private initiatives endorsed by the city flourish (CitiBikes, Living Complexes, etc.). These aren't only inaccessible monetarily by the residents, but they

incentivize and encourage our expulsion from our homes, the drowning of local businesses, and harm our purchasing power. (on the right) We see 1: business chains arriving to the neighborhood. 2. A CitiBike station. 3. A resident's bicycle. 4. A street vendor's yellow stand. 5. New living complex. This corner always had small businesses, representing the different cultures of [the community's] residents, with bakeries or food stands but we see how large chains have arrived or are arriving to modify and impose gentrification. We constantly learn about new families who cannot pay their rent and how they are displaced to other communities further away because of this predatory and lapidary model that uproots and exposes their vulnerability as a migrant community.

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
