# Peer review of "¿Dónde Vive la Ciencia en su Comunidad?: How a Community Is Using Photovoice to Reclaim Local Green Spaces"

_socsci, doi:10.3390/socsci14010013_

Round 1

Reviewer 1 Report

Comments and Suggestions for Authors

see attached file

Author Response

Thank you so much for your thoughtful questions, and for inviting us to think more deeply about the limitations of our sample size. We deeply appreciate the time, energy, and attention to detail that you provided us, and are appreciative of your feedback, which we know will serve to strengthen our paper. We have incorporated most of the suggestions you made. Those changes are highlighted throughout the manuscript. Please see below, in blue, a point-by-point response to your comments. All page numbers refer to the revised manuscript file, which includes the incorporated changes. 

Reviewer 1

Review for:

¿Dónde Vive la Ciencia en su Comunidad?: How a community is using photovoice to

reclaim local green spaces

Thank you – this was a well-written and enjoyable paper. The article examines the use of community-based participatory research to highlight issues of gentrification and community empowerment, thereby addressing social and environmental justice for communities. The paper argues that this participatory research model offers opportunities to include the voices of the community in policy and research.

Thank you! We’re glad that this was clear in the manuscript. I have two areas for the authors to think about and add further explanation:

- Given the exceptionally small numbers of community researchers who engaged in this

project, how can they claim to represent in any meaningful way the wider community? I think that this needs further defending in the paper.

- Secondly, and related to the above, the authors tends to homogenise ‘the’ community

throughout the paper. I am sure that a community of 2.3m people has many different

views on urban planning, neighbourhood redesign etc, and does not speak with a single

voice. What gives the (few) community members who participated as community

researchers (from the Latine community) the legitimacy to speak for ‘the’ wider

community or communities who inhabit these spaces and presumably have a similar

right to have their voices and views heard? Who will facilitate disputes between different

groups if and when they have competing visions for their future? E.g. between young and old as well as different ethnic groups. 

In response to both points -- Agreed! We believe that the process of adapting photovoice to be culturally responsive is one that could be relevant across various contexts; however, we recognize that the small sample size means that the findings are not generalizable. We’ve added context in the methods section (p. 7), as well as a section on limitations in the discussion (p. 29) 

I do not consider these points to require major revisions. As I say on the whole this is a very good paper however I feel it makes several points about empowerment and social justice that have been made many times before. It would be nice to see some more concrete examples of just exactly HOW communities, with the aid of educators, allies and compas, can engage withpolicy making for change within their communities.

Thank you! We’ve added more context in the discussion (pp. 24-28).

Specific comments/typos:

Abstract – spell out the STEM acronym the first time you use this, it helps the reader to know

exactly what you mean. After that you can use the acronym. Updated!

P.9 – typo in ‘CPBR’ – it should be CBPR – community-based participatory research. Updated!

P.11 – the photo ‘group of people in a classroom’ already appears above. Can you replace it with a different one or just have two photos here rather than three? With limited space I don’t think you can justify putting the same photo in twice. New images included

P21 – is there a citation missing for Bronfenbrenner’s socio-ecological model? Yes! Updated

P.23 – is there a typo in the last two sentences of section 4.1.2? It looks like two sentences but

should be one, with a comma between ‘well-being’ and ‘funding’ rather than a full stop.Thank you! Updated.

Reviewer 2 Report

Comments and Suggestions for Authors

Article Title:

¿Dónde Vive la Ciencia en su Comunidad?: How a community is using photovoice to reclaim local green spaces

Reviewer Comments:

The work presented in this manuscript is significant and valuable in times that of heightened racist nativism. The authors have endeavored to pursue a very ambitious project with implications to support and advance epistemic justice and wellbeing/healing, as well as environmental justice, which I commend them for. There are some ideas that, in my humble opinion, can be further clarified and strengthened – or are in need for further development. I have offered some comments in particular sections of the paper where these can be further elaborated on or clarified. 

Below, I offer few remarks and reflections, which the authors may wish to respond to and address in their revision/resubmission process: 

·      Define “decoloniality” and what is mean by “decolonial strategies.” Where there efforts to engage with local Indigenous communities and affirm their sovereignty over land-rights? The issue of gentrification is very complex especially as it relates to land rights and sovereignty. The issue is further complicated by patterns of gentrification and displacement and reinforce cycles of who get to be/live and thus occupy certain lands. How where these tensions, if at all, engaged by the community, researchers, collaborators?

·      Cite: Eve Tuck’s work on “refusing deficit-based research”; Andrew Jolívette’s “research justice” framework; consider highlighting the work of Latiné scholars that have used photovoice; Torre & Ayala’s work on “PAR: Entremundos” also seems relevant and important to mention

·      What does it mean to relinquish titles of “researcher”? How does one navigate the complexities of being positioned in such ways? How does one ethically trouble the power d dynamics that surface in PAR, or community-engaged research? Here, I’m especially thinking about the work of Maria Elena Torre’s concept of nos-otras and choques, referencing the work of Gloria E. Anzaldúa

·      Consider elaborating or providing some examples for how reflexivity was facilitated, especially as it is a fundamental premise of PAR and community-engage research, and one that would likely support the development of humanizing and justice-oriented projects with an environmental justice focus oriented toward mujerista and liberatory praxes.

·      Having presented and discussed such powerful photo-narratives via the outcomes of the photovoice project engaged, the authors purport that group members identified opportunities for generating community-driven change and the fostering of spaces toward healing and growth. I wonder what this means or looks like? What actions or steps have community researchers engaged in toward this process of healing and repair/restoring wellbeing in the community, especially for those who are impacted by inequitable systems? For example, what cultural and intergenerational traditions have community researchers sough to facilitate to restore justice and health/wellness in the community and land? And, how do these “interventions” or praxes align with an Indigenous Ecological Systems Model as per Fish & Syed’s model?

·      The closing sections of the paper are quiet strong and well-articulated. However, I would encourage the authors to consider elaborating on the implications or recommendations for policymakers because it seems that many of the other recommendations hinge to a degree on the capacity for policies and institutional actors to leverage their power and resources in alignment with voices and agency of communities, community researchers and practitioners, including educators. How can policymakers support community-driven development and policies? Are these examples form the study featured that demonstrate this that can further help illustrate what this looks like in practice?

·      Furthermore, and in relation to the above point, what does it mean for policymakers to promote STEM education? How can STEM education – as well as the arts, which this study is a prime example of the power of art via photography – further support community change and environmental justice on the terms and conditions of communities themselves? What avenues can policymakers embrace to integrate and actually respond to community input and voice in policy development and change?

Comments on the Quality of English Language

Author Response

We are so grateful for the great care that you took in writing your review for us! Thank you for your time, energy, and generosity in sharing relevant resources that have pushed our thinking (and inspired us!). Thank you also for your thought-provoking questions, which invited more dialogue that we know will only serve to strengthen the paper. We have incorporated most of the suggestions you made. Those changes are highlighted throughout the manuscript. Please see below, in blue, a point-by-point response to your comments. All page numbers refer to the revised manuscript file, which includes the incorporated changes. 

Reviewer 3

The work presented in this manuscript is significant and valuable in times that of heightened racist nativism. The authors have endeavored to pursue a very ambitious project with implications to support and advance epistemic justice and wellbeing/healing, as well as environmental justice, which I commend them for. 

Thank you!

There are some ideas that, in my humble opinion, can be further clarified and strengthened – or are in need for further development. I have offered some comments in particular sections of the paper where these can be further elaborated on or clarified. 

 We appreciate your insights.

Below, I offer few remarks and reflections, which the authors may wish to respond to and address in their revision/resubmission process: 

  •     Define “decoloniality” and what is mean by “decolonial strategies.” Where there efforts to engage with local Indigenous communities and affirm their sovereignty over land-rights? The issue of gentrification is very complex especially as it relates to land rights and sovereignty. The issue is further complicated by patterns of gentrification and displacement and reinforce cycles of who get to be/live and thus occupy certain lands. How where these tensions, if at all, engaged by the community, researchers, collaborators?

Thank you. This is so important, and the questions you brought up are ones we are still contending with as a group. We have included our definitions of decolonial strategies, based on Mignolo and Quijano’s work on p. 2 and pp. 4-6. Additionally, we attempt to locate epistemic disobedience in this larger project of decolonization in the discussion on pp. 24-26. 

  •     Cite: Eve Tuck’s work on “refusing deficit-based research”; Thank you – we really appreciated Tuck’s framing of “desire-based research” (included on p. 25)  Andrew Jolívette’s “research justice” framework Such a helpful frame! (included on pp. 2 & 3); consider highlighting the work of Latiné scholars that have used photovoice; Torre & Ayala’s work on “PAR: Entremundos” also seems relevant and important to mention Thank you, again – very helpful in strengthening the manuscript (included on pp. 2, 6, & 14).
  •     What does it mean to relinquish titles of “researcher”? How does one navigate the complexities of being positioned in such ways? How does one ethically trouble the power d dynamics that surface in PAR, or community-engaged research? Here, I’m especially thinking about the work of Maria Elena Torre’s concept of nos-otras and choques, referencing the work of Gloria E. Anzaldúa Included both 

Agreed – Torre’s analysis is helpful, as well as Jolivette’s. We’ve added more context on pp. 3-4

  •     Consider elaborating or providing some examples for how reflexivity was facilitated, especially as it is a fundamental premise of PAR and community-engage research, and one that would likely support the development of humanizing and justice-oriented projects with an environmental justice focus oriented toward mujerista and liberatory praxes.

Added some more context on reflexive nature of the data analysis on p. 10.

  •     Having presented and discussed such powerful photo-narratives via the outcomes of the photovoice project engaged, the authors purport that group members identified opportunities for generating community-driven change and the fostering of spaces toward healing and growth. I wonder what this means or looks like? What actions or steps have community researchers engaged in toward this process of healing and repair/restoring wellbeing in the community, especially for those who are impacted by inequitable systems? For example, what cultural and intergenerational traditions have community researchers sough to facilitate to restore justice and health/wellness in the community and land? And, how do these “interventions” or praxes align with an Indigenous Ecological Systems Model as per Fish & Syed’s model? 

We honor that this work is ongoing (and still very much in its seedling phase). We’ve shared some of the “spirit-work” we’ve engaged in on pp. 4, 25, and 28. We’ve also elaborated on how we are sharing this work with broader communities at the 2 institutions, and how this sharing has influenced institutional practices at both institutions on p. 25.

  •     The closing sections of the paper are quiet strong and well-articulated. However, I would encourage the authors to consider elaborating on the implications or recommendations for policymakers because it seems that many of the other recommendations hinge to a degree on the capacity for policies and institutional actors to leverage their power and resources in alignment with voices and agency of communities, community researchers and practitioners, including educators. How can policymakers support community-driven development and policies? Are these examples form the study featured that demonstrate this that can further help illustrate what this looks like in practice?
  •     Furthermore, and in relation to the above point, what does it mean for policymakers to promote STEM education? How can STEM education – as well as the arts, which this study is a prime example of the power of art via photography – further support community change and environmental justice on the terms and conditions of communities themselves? What avenues can policymakers embrace to integrate and actually respond to community input and voice in policy development and change? 

In response to th last two comments – Thank you for this guidance – we’ve further developed our recommendations for policymakers in response to your questions (pp. 26-30).